# Conditional Representation Learning
# for Customized Tasks

**Honglin Liu**[1]**, Chao Sun**[2]**, Peng Hu**[1]**, Yunfan Li**[1]*, **Xi Peng**[1,3]*

[1]College of Computer Science, Sichuan University, Chengdu, China
[2]Aerospace Information Research Institute, Chinese Academy of Sciences, Beijing, China
[3]National Key Laboratory of Fundamental Algorithms and Models
for Engineering Numerical Simulation, Sichuan University, Chengdu, China
{TristanLiuHL, penghu.ml, yunfanli.gm, pengx.gm}@gmail.com
{sunchao}@aircas.ac.cn

## Abstract

Conventional representation learning methods learn a universal representation that primarily captures dominant semantics, which may not always align with customized downstream tasks. For instance, in animal habitat analysis, researchers prioritize scene-related features, whereas universal embeddings emphasize categorical semantics, leading to suboptimal results. As a solution, existing approaches resort to supervised fine-tuning, which however incurs high computational and annotation costs. In this paper, we propose Conditional Representation Learning (CRL), aiming to extract representations tailored to arbitrary user-specified criteria. Specifically, we reveal that the semantics of a space are determined by its basis, thereby enabling a set of descriptive words to approximate the basis for a customized feature space. Building upon this insight, given a user-specified criterion, CRL first employs a large language model (LLM) to generate descriptive texts to construct the semantic basis, then projects the image representation into this conditional feature space leveraging a vision-language model (VLM). The conditional representation better captures semantics for the specific criterion, which could be utilized for multiple customized tasks. Extensive experiments on classification and retrieval tasks demonstrate the superiority and generality of the proposed CRL. The code is available at XLearning-SCU/2025-NeurIPS-CRL.

## 1 Introduction

Representation learning aims at extracting meaningful patterns from raw data to create representations that are easier to understand and process. Its impact spans a wide range of downstream tasks, such as classification and retrieval. In classification, representation learning enhances the discrimination and linear separability of features, significantly improving performance across diverse data modalities, including images [29], text [41], and video [59]. Similarly, in retrieval tasks, representation learning underpins efficient and accurate query-to-item matching, as evidenced by developments in image retrieval [18] and cross-modal retrieval [50]. In recent years, driven by self-supervision techniques such as contrastive learning [6, 23, 19, 7, 71] and mask prediction [9, 22, 73, 61], representation learning methods have undergone rapid advancements, leading to substantial performance improvements across various fields, including graph [42], point-cloud [64], and skeleton [65].

Though remarkable progress has been made, a crucial yet often overlooked question remains: **What underlying criterion governs the learned representation?** In fact, most existing representation learning methods inherently impose an implicit criterion. Previous research [56] has demonstrated

---

*Corresponding Authors.

39th Conference on Neural Information Processing Systems (NeurIPS 2025).

that representations learned by existing approaches exhibit a strong bias toward a single dominant aspect, typically "shape" or "category"—as these are the most salient features in many datasets. This inherent bias causes models to prioritize specific attributes while disregarding other potentially informative features, such as "texture" and "color". Consequently, the resulting universal embeddings predominantly capture a single prominent criterion, leading to sub-optimal performance in downstream tasks that rely on alternative perspectives. As illustrated in Fig. 1, existing methods primarily identify the elephant "category", which is insufficient for customized tasks like population monitoring or habitat analysis. In comparison, our CRL could adaptively capture "count" and "scene" semantics, demonstrating broader generality. This narrow focus ultimately constrains the generalization capability of representation learning methods, underscoring the need for more adaptable and criterion-aware approaches.

To transform the image representation to align with specific criteria, a straightforward approach would be supervised fine-tuning [16, 35], where models are retrained using labeled data that adhere to the given criterion. However, such a paradigm is not always practical due to the substantial annotation effort required. In the unsupervised scenario, where only images and a user-specified criterion are provided, a feasible solution is to query visual question answering (VQA) models [52, 69, 31] to extract relevant attributes from each image. However, this approach is computationally expensive and requires additional representation learning steps for the generated textual responses. With these considerations, an efficient way of learning the criterion-oriented image representation is highly expected.

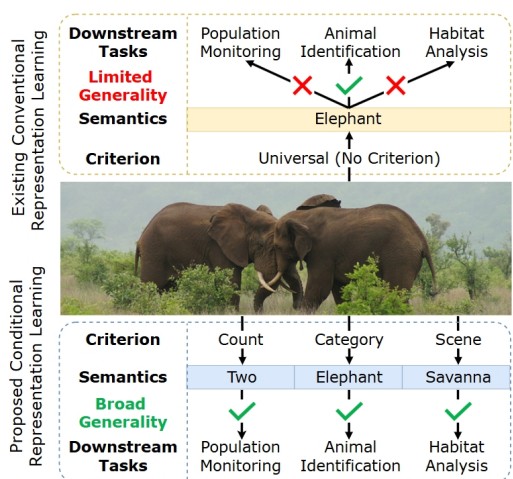

Figure 1: Existing conventional representation learning learns a universal representation that prioritizes the dominant semantics while overlooking other meaningful features, limiting their adaptability to customized tasks. In contrast, our proposed conditional representation learning (CRL) extracts representations conditioned on specific criteria, enhancing its applicability.

In recent years, researchers have also been exploring computationally efficient approaches to learning useful representations. Goal-conditioned works [46, 43] target learning representations that meet the required outcomes or goal states. An area that is more closely related to our work is task-conditioned works [70, 2], which aim to learns representations that reveal the underlying correlations among different tasks. For example, taskonomy [70] computes the optimal transfer learning paths among tasks (point matching, reshading, etc.) to minimize the amount of required annotation. While there are certain commonalities between these works and ours, they haven't investigated the relationship between criteria and representations.

In this paper, we introduce Conditional Representation Learning (CRL), a novel approach that adapts the image representation to any user-specified criterion. Unlike conventional representation learning methods, which primarily focus on general-purpose feature extraction, CRL constructs a customized feature space by leveraging the concept of basis transformation. The key insight behind CRL is that the semantics of a feature space are determined by its basis. For example, in a three-dimensional Cartesian coordinate system, the x, y, and z unit vectors define the space, allowing for the decomposition of any vector. Similarly, in color theory, red, green, and blue serve as the basis for the trichromatic color space, enabling the synthesis of all perceivable hues. Extending this idea to high-dimensional semantic representations, a well-chosen set of descriptive words can form a basis for a customized feature space, which captures specific semantic properties aligned with a user-defined criterion. Building on this perspective, CRL formulates conditional representation learning as a basis transformation process. Given a user-specified criterion, we first employ a large language model (LLM) to generate a set of descriptive texts that serve as a semantic basis, spanning the relevant feature space. We then utilize a vision-language model (VLM) to encode both the generated texts and the images, obtaining their representations respectively. Finally, we project the image representation into the conditional feature space with the textual representation acting as a

basis. The transformed conditional representation would be more expressive under the specified criterion, which could be utilized for downstream tasks that require customized semantics.

The major contributions of this paper could be summarized as follows:

- Different from conventional representation learning that primarily captures a single dominant semantics, we propose conditional representation learning (CRL), which enables learning representations tailored to arbitrary user-specified criteria.

- We formulate CRL as a basis transformation process, offering a computationally efficient and highly generalizable solution. It eliminates the reliance on supervised fine-tuning while substantially improving the applicability and interpretability of the learned representation.

- Extensive experiments validate the effectiveness and generality of CRL in customized classification and retrieval, showcasing its superiority in seamlessly adapting to varying criteria and tasks.

## 2   Related Work

### 2.1   Representaion Learning

Representation learning aims to extract informative features from raw data, facilitating downstream tasks like classification and retrieval. As a classic method, autoencoder [24] learns compact representations through unsupervised reconstruction. Building upon it, denoising autoencoders [58] and variational autoencoders [27] have been proposed to enhance the robustness and structure of the learned latent representations. In the past few years, the field has further evolved with self-supervised learning techniques, which encourage models to learn semantical features by addressing pretext tasks such as patch and rotation prediction [10, 17], solving jigsaw puzzles [44], and colorization [72]. A notable advancement in this direction is contrastive learning, exemplified by methods like SimCLR [6] and MoCo [23], which leverage instance discrimination to learn discriminative representations. More recently, the emergence of large language models (LLMs) such as GPT [5] and vision-language models (VLMs) like CLIP [48] has introduced a more interpretable approach for representation learning. A series of works [74, 15, 47, 38, 21] have then researched using CLIP to improve zero-shot or few-shot image classification performance. By analyzing the Vision Transformer [13] architecture of CLIP, studies such as Text-Span [14] have shed light on the underlying semantics captured by individual attention heads. Leveraging the strengths of LLMs and VLMs, approaches like VCD [40], LaBo [66] and LM4CV [63] have demonstrated that interpretable representation learning can achieve performance on par with black-box methods in downstream image classification.

Despite significant progress, most existing representation learning approaches remain centered on a single criterion, typically "category" or "shape", while overlooking other meaningful semantic dimensions. This narrow focus limits the generalizability of learned representations, often necessitating extensive supervised fine-tuning when adapting to tasks that depend on alternative semantic cues. To address this limitation, we advocate for a paradigm shift from universal to conditional representation learning, an underexplored yet promising direction. Specifically, our approach first constructs a semantic basis composed of descriptive texts aligned with a user-specified criterion. Leveraging this customized basis, we transform the image representation to enable conditional adaptation, enhancing the flexibility and applicability of learned features without additional laborious fine-tuning.

### 2.2   Conditional Similarity

Conditional similarity refers to the similarity between samples based on specific criteria. This concept was first formalized by CSN [57], which learns multiple feature spaces to enable customized fashion item retrieval under different criteria. With the advent of representation learning, a series of tailored fashion retrieval approaches have been developed [39, 11, 12], significantly improving the retrieval performance. Recently, the idea of conditional similarity has gained traction in the clustering domain [37]. Driven by the powerful language processing capabilities of large-scale pre-trained models, IC|TC [28] pioneers the concept of customized clustering by directly querying VLMs and LLMs to obtain clustering results based on specific criteria. However, this approach incurs high computational costs. To address this limitation, Multi-Map [68] introduces a more cost-efficient alternative, injecting customized semantics from VLM and the LLM to guide the clustering process.

Despite the success of existing methods, they are all delicately designed for specific tasks, limiting their generalization ability to other domains. In contrast, we propose CRL, a simple yet effective method for learning general conditional representation, which could seamlessly adapt to diverse customized tasks.

## 3 Method

This section details the proposed Conditional Representation Learning (CRL) framework, which consists of basis construction and representation transformation. As depicted in Fig. 2, given a user-specified criterion, CRL first constructs a customized basis by querying an LLM about descriptive words. Subsequently, CRL computes the conditional image representation through a basis transformation operation.

### 3.1 Basis Construction

Mathematically, a basis refers to a set of linearly independent vectors[2] that span the entire space. For example, in the three-dimensional Cartesian coordinate system, vectors $(1, 0, 0)$, $(0, 1, 0)$, and $(0, 0, 1)$, which denote the x, y, and z axes, form a basis since any vector in the space can be expressed as a linear combination of these three vectors. Analogously, in the trichromatic color space, "red", "green", and "blue" form a basis as they could compose all possible hues. From a broader view, a set of descriptive words related to the user-specified criterion, that spans the customized feature space, intrinsically acts as the basis as well.

To construct the basis under the specific criterion $C$, we query an LLM to generate the related descriptive texts $W$ via

$$W = \text{LLM}(P_1, C), \tag{1}$$

where $P_1$ denotes the LLM prompt template. As a general solution, we use the following prompt for all customized tasks:

Generate common expressions to describe the $C$, as many as possible.

where $C$ is replaced with the user-specified criterion words such as "color", "shape", "texture", etc. Notably, we incorporate additional instructions to encourage the LLM to produce formatted, comprehensive texts and avoid repetitions, which are detailed in the Appendix.

Given the prompted query, the LLM would generate texts $W$ semantically correlated with the user-specified criterion, transforming the abstract criterion into a concrete textual basis. Once the descriptive texts $W$ are obtained, we feed them into a VLM text encoder $\text{VLM}_{\text{text}}$ to compute their normalized representation $\mathbf{T}$ via

$$\mathbf{T} = \text{VLM}_{\text{text}}(P_2, C, W), \tag{2}$$

where $P_2$ denotes the VLM prompt constructed as follows:

Objects with the $C$ of $W$.

It is worth noting that, when prior knowledge about the dataset is available, the word "Objects" could be replaced by more specific descriptions. The complete prompts used for all customized tasks in this paper, as well as the LLM responses, are attached in the Appendix.

As previously discussed, the text representation $\mathbf{T}$ could act as the basis spanning the customized feature space. Remarkably, compared with the basis of the classic universal feature space, the constructed basis $\mathbf{T}$ enjoys superior interpretability where each dimension has an explicit physical meaning.

### 3.2 Representation Transformation

After acquiring the text basis, we leverage it to transform the universal representation into the conditional representation, by projecting data into the constructed customized feature space.

---

[2]In this paper, we relax the linear independence requirement and allow redundancy in the constructed basis.

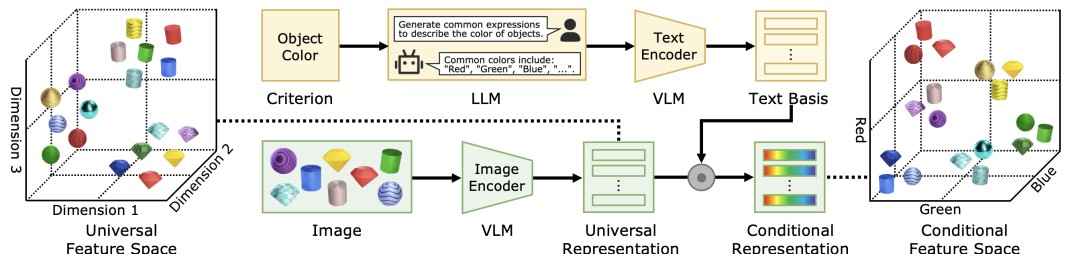

Figure 2: The overall framework of the proposed CRL. Given images and a user-specified criterion (*e.g.*, "color"), CRL first queries an LLM to generate descriptive texts semantically related to the criterion (*e.g.*, "red", "green" and "blue"). Then, CRL encodes the generated texts and original images through a VLM. Subsequently, CRL projects the original image representation (*e.g.*, dominated by "shape") into the conditional feature space spanned by the textual representation. The transformed conditional representation would be more expressive under the specified criterion and enjoy superior interpretability, facilitating customized downstream tasks.

To be specific, we first feed the images $\mathbf{X}$ into the VLM image encoder $\text{VLM}_{\text{image}}$ to obtain their normalized representation $\mathbf{I}$ via

$$\mathbf{I} = \text{VLM}_{\text{image}}(\mathbf{X}). \tag{3}$$

Subsequently, we transform the image representation by projecting it to the customized space spanned by text basis $\mathbf{T}$, namely,

$$\mathbf{R} = \mathbf{I}\mathbf{T}^\top, \tag{4}$$

where $\mathbf{R}$ denotes the transformed conditional representation. The validity of this transformation exploits the alignment between image and text modalities in the VLM's feature space. The conditional representation $\mathbf{R}$ emphasizes the attributes related to the user-specified criterion, and thus is more favorable in customized tasks.

The complete process of our CRL is outlined in Algorithm 1. To deliver a more intuitive understanding of CRL's working mechanism and underlying rationale, we provide an example about learning a color-conditioned representation as illustrated in Fig. 2.

Consider the customized clustering task, which aims at grouping images based on their colors. The original image representation is dominated by the most significant shape information, which is suboptimal for color-based grouping. To build a customized feature space focusing on colors, we first query an LLM about the common colors. Supposing the LLM outputs descriptive texts $W = \{$"red", "green", "blue"$\}$, we calculate the text basis as

$$\mathbf{T} = [t_1^\top, t_2^\top, t_3^\top]^\top, \tag{5}$$

where $\{t_1, t_2, t_3\}$ denote the rows of $\mathbf{T}$, corresponding to the representations of "red", "green", and "blue".

Then we project the $k$-th original image representation $i_k$ to conditional representation $r_k$ via

$$r_k = i_k \mathbf{T}^\top = [i_k \cdot t_1, i_k \cdot t_2, i_k \cdot t_3]. \tag{6}$$

As shown in Eq. (6), the transformed conditional representation of the $k$-th image refers to the projection of its original representation onto the text basis $\mathbf{T}$. Consequently, the three elements of $r_k$ correspond to its degree of "red", "green", and "blue", respectively. In other words, $r_k$ is more expressive than $i_k$ under the "color" criterion, leading to superior performance on the customized clustering task.

## 4 Experiments

To assess the conditional representation learning performance of the proposed CRL, we apply it to two classic downstream tasks, including classification and retrieval. Notably, different from standard

**Algorithm 1** Conditional Representation Learning (CRL)
___
**Input:** Criterion $C$, LLM Prompt $P_1$, VLM Prompt $P_2$, Images $\mathbf{X}$
**Output:** Transformed Conditional Representation $\mathbf{R}$
 1: Query an LLM to generate the descriptive texts $W$ related to the user-specified criterion $C$ via Eq.(1).
 2: Compute the text basis $\mathbf{T}$ via Eq.(2).
 3: Compute the original universal image representation $\mathbf{I}$ via Eq.(3).
 4: Transform $\mathbf{I}$ into conditional representation $\mathbf{R}$ via Eq.(4), which could be then utilized for various customized tasks.
___

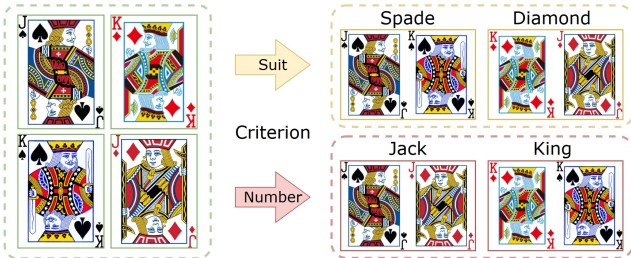

Figure 3: A customized classification example of classifying poker cards based on the criteria of "suit" and "number", respectively.

representation learning, CRL focuses on learning conditional representation, and thus the downstream classification and retrieval are based on various customized criteria. After that, parameter analysis is conducted to investigate the robustness of CRL.

## 4.1 Customized Classification

As shown in Fig. 3, customized classification aims to classify samples into different semantic categories under the specific criterion, which includes two subtasks, *i.e.*, supervised few-shot learning and unsupervised clustering.

### 4.1.1 Customized Few-shot Learning

**Dataset.** For this task, we utilize Clevr4-10k [56] and Cards [67] as benchmark datasets. Clevr4-10k is a synthetic dataset consisting of $10,531$ samples and $4$ distinct data partition criteria, categorized by "shape", "texture", "color", and "count", respectively. Cards is a poker card dataset comprising $8,029$ samples, organized according to $2$ criteria, *i.e.*, "number" and "suit".

**Setup.** For fair comparisons, we adopt the logistic regression function from the scikit-learn package [45] to perform few-shot learning, under the number of shots $1, 5, 10$ per class, respectively. To alleviate the influence of randomness, we stochastically select the training data 20 times for each shot and report the mean result. As for the backbone, we adopt ViT-B/32 pre-trained on CLIP, keeping the same with Section 4.1.2.

**Metric.** For the task of customized few-shot learning, we adopt accuracy (ACC) as the evaluation metric.

**Baseline.** We conduct comparisons between proposed CRL and image representations of CLIP [48], ALIGN [25] and MetaCLIP [62] across six semantic criteria.

**Performance.** As illustrated in Table. 1, CRL achieves a noticeable improvement over CLIP, ALIGN and MetaCLIP across most experimental settings, with a mean accuracy gain of nearly $10\%$. Particularly, CRL gains significant improvements when the target criterion differs substantially from the originally dominant one, such as 'color' (nearly $+40\%$ at 1-shot). The consistent performance advantage indicates that CRL's representation exhibits a better generality under multiple criteria.

Table 1: Performance on the task of customized few-shot learning.

| Method | Clevr4-10k | | | | | | | | | Mean |
|---|---|---|---|---|---|---|---|---|---|---|
| | Texture | | | Shape | | | Color | | | |
| | 1 | 5 | 10 | 1 | 5 | 10 | 1 | 5 | 10 | |
| CLIP [48] | 17.46 | 29.39 | 36.26 | 58.16 | 83.17 | 89.47 | 26.85 | 57.33 | 70.00 | 52.01 |
| ALIGN [25] | 18.80 | 34.35 | 45.22 | **73.40** | 91.82 | 95.02 | 20.08 | 41.89 | 56.45 | 53.00 |
| MetaCLIP [62] | 17.68 | 30.96 | 39.03 | 70.13 | 91.69 | 95.47 | 22.37 | 46.71 | 61.74 | 52.86 |
| BLIP2 [30] | 15.93 | 25.23 | 32.58 | 72.91 | **95.18** | 97.88 | 28.96 | 60.53 | 73.25 | 55.83 |
| **CLIP+CRL** | 18.76 | 35.54 | 45.54 | 58.67 | 86.61 | 92.29 | **65.28** | **88.89** | **93.08** | 64.96 |
| **ALIGN+CRL** | **20.91** | **41.77** | **54.92** | 63.05 | 92.74 | 96.25 | 60.26 | 87.38 | 92.56 | **67.76** |
| **MetaCLIP+CRL** | 18.14 | 34.89 | 44.69 | 66.36 | 92.01 | 92.50 | 62.41 | 88.45 | 92.50 | 66.11 |
| **BLIP2+CRL** | 16.35 | 34.67 | 47.28 | 73.22 | 95.12 | **97.90** | 63.75 | 86.16 | 92.13 | 67.40 |

| Method | Clevr4-10k | | | Cards | | | | | | Mean |
|---|---|---|---|---|---|---|---|---|---|---|
| | Count | | | Number | | | Suits | | | |
| | 1 | 5 | 10 | 1 | 5 | 10 | 1 | 5 | 10 | |
| CLIP [48] | 17.50 | 23.43 | 25.45 | 20.63 | 33.73 | 41.84 | 37.65 | 56.36 | 65.98 | 35.84 |
| ALIGN [25] | 14.64 | 21.63 | 25.16 | 16.97 | 24.70 | 29.15 | 34.67 | 52.75 | 61.78 | 31.27 |
| MetaCLIP [62] | 16.61 | 22.64 | 24.92 | 37.47 | 55.03 | 65.16 | 20.71 | 35.16 | 42.97 | 35.63 |
| BLIP2 [30] | 16.92 | 25.63 | 29.38 | 27.21 | 45.54 | 55.94 | 44.61 | 70.14 | 78.16 | 43.73 |
| **CLIP+CRL** | **23.38** | 29.59 | 32.40 | 17.66 | 44.52 | 51.09 | 37.10 | 67.16 | 72.64 | 41.73 |
| **ALIGN+CRL** | 18.16 | 32.62 | 36.80 | 17.39 | 30.61 | 35.93 | 42.13 | 76.36 | 80.11 | 41.12 |
| **MetaCLIP+CRL** | 17.36 | 26.29 | 29.93 | **42.32** | **71.88** | **77.32** | 25.30 | 50.53 | 56.90 | 44.20 |
| **BLIP2+CRL** | 23.06 | **34.86** | **39.07** | 23.47 | 61.19 | 70.05 | **49.57** | **80.44** | **84.06** | **51.75** |

Table 2: Performance on the task of customized clustering.

| Method | Clevr4-10k | | | | | | | | | Mean |
|---|---|---|---|---|---|---|---|---|---|---|
| | Texture | | | Shape | | | Color | | | |
| | NMI | ACC | ARI | NMI | ACC | ARI | NMI | ACC | ARI | |
| CC [32] | 0.16 | 11.34 | 0.00 | **94.66** | **96.89** | **93.90** | 16.54 | 11.42 | 0.07 | 36.11 |
| SCAN [54] | 0.41 | 11.97 | 0.86 | 90.99 | 89.10 | 84.03 | 0.20 | 11.51 | 0.01 | 32.12 |
| Multi-Map [68] | 3.77 | 17.25 | 1.81 | 67.48 | 66.01 | 57.40 | 56.83 | 56.46 | 45.73 | 41.42 |
| CLIP [48] | 1.11 | 13.09 | 0.41 | 74.22 | 73.19 | 64.15 | 0.83 | 12.23 | 0.27 | 26.61 |
| ALIGN [25] | 1.36 | 13.30 | 0.41 | 89.33 | 86.77 | 83.37 | 0.47 | 11.79 | 0.10 | 31.88 |
| MetaCLIP [62] | 1.44 | 12.75 | 0.42 | 80.54 | 77.17 | 71.58 | 0.32 | 11.85 | 0.06 | 28.46 |
| BLIP2 [30] | 0.79 | 12.32 | 0.28 | 86.98 | 85.68 | 81.17 | 0.99 | 11.92 | 0.24 | 31.15 |
| **CLIP+CRL** | 10.74 | 25.11 | 6.35 | 78.69 | 83.05 | 72.42 | **88.67** | **88.05** | **82.30** | 59.49 |
| **ALIGN+CRL** | **15.08** | **26.08** | **9.18** | 88.27 | 87.63 | 81.83 | 85.07 | 76.15 | 72.69 | 60.22 |
| **MetaCLIP+CRL** | 12.74 | 25.89 | 7.28 | 87.32 | 88.15 | 82.98 | 88.35 | 86.27 | 81.08 | **62.23** |
| **BLIP2+CRL** | 6.46 | 18.77 | 3.37 | 90.11 | 88.91 | 84.52 | 84.67 | 81.97 | 74.85 | 59.29 |

| Method | Clevr4-10k | | | Cards | | | | | | Mean |
|---|---|---|---|---|---|---|---|---|---|---|
| | Count | | | Number | | | Suits | | | |
| | NMI | ACC | ARI | NMI | ACC | ARI | NMI | ACC | ARI | |
| CC [32] | 2.08 | 14.67 | 1.09 | 24.91 | 26.34 | 12.30 | 24.94 | 39.21 | 16.87 | 18.05 |
| SCAN [54] | 3.42 | 14.29 | 1.23 | 11.11 | 18.21 | 17.60 | 15.01 | 32.02 | 9.48 | 13.60 |
| Multi-Map [68] | 11.38 | 20.13 | 7.67 | 16.32 | 20.61 | 7.95 | 14.02 | 46.65 | 11.08 | 17.31 |
| CLIP [48] | 9.50 | 19.02 | 5.70 | 16.84 | 18.91 | 8.44 | 16.52 | 43.74 | 12.93 | 16.84 |
| ALIGN [25] | 0.63 | 12.50 | 0.19 | 14.86 | 17.51 | 6.47 | 3.49 | 31.72 | 2.31 | 9.96 |
| MetaCLIP [62] | 7.62 | 17.27 | 3.97 | 17.39 | 19.78 | 9.04 | 15.48 | 38.72 | 13.11 | 15.82 |
| BLIP2 [30] | 6.11 | 16.36 | 3.13 | 24.34 | 25.25 | 13.08 | 31.26 | 47.04 | 22.25 | 20.98 |
| **CLIP+CRL** | 25.57 | 26.24 | 12.54 | 24.79 | 28.19 | 12.14 | 39.71 | 67.15 | 37.59 | 30.44 |
| **ALIGN+CRL** | 22.78 | 26.59 | 12.18 | 20.12 | 25.32 | 10.24 | 42.94 | 44.47 | 27.27 | 27.27 |
| **MetaCLIP+CRL** | 12.22 | 20.80 | 6.18 | 39.07 | 41.63 | 24.37 | 45.19 | 58.71 | 36.97 | 31.68 |
| **BLIP2+CRL** | **28.55** | **30.92** | **16.28** | **46.55** | **48.35** | **32.31** | **60.86** | **76.07** | **55.94** | **43.98** |

### 4.1.2 Customized Clustering

**Dataset.** We continue to perform experiments on Clevr4-10k and Cards datasets for the task of customized clustering.

**Setup.** We directly conduct k-means on the representations obtained by CRL to get the clustering. Keeping the same as the customized few-shot learning, we also perform k-means 20 times and report the average clustering result. As for the backbone, we follow the previous method [68], adopting ViT-B/32 pre-trained on CLIP.

**Metric.** Three widely used clustering metrics, namely Normalized Mutual Information (NMI), Accuracy (ACC), and Adjusted Rand Index (ARI), are used for evaluation. Higher scores indicate better clustering results.

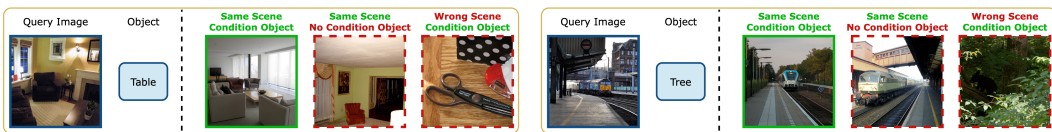

(a) Focus on an object           (b) Change an object

Figure 4: Two settings of the customized similarity retrieval task.

Table 3: Performance on the task of customized similarity retrieval. The symbol * means using the fine-tuned CLIP model weights.

| Method | Focus | | | Change | | | Mean |
|---|---|---|---|---|---|---|---|
| | R@1 | R@2 | R@3 | R@1 | R@2 | R@3 | |
| $CLIP_{image}$ | 9.4 | 17.0 | 25.4 | 7.6 | 17.1 | 25.5 | 17.0 |
| $CLIP_{text}$ | 7.4 | 14.0 | 23.0 | 8.1 | 16.4 | 24.7 | 15.6 |
| $CLIP_{image+text}$ | 11.5 | 20.1 | 29.2 | 9.8 | 20.0 | 28.9 | 19.9 |
| Pic2Word [49] | 9.9 | 19.3 | 27.4 | 8.6 | 18.2 | 26.1 | 18.3 |
| SEARLE [4] | 10.8 | 18.2 | 27.9 | 8.3 | 15.6 | 25.8 | 17.8 |
| LinCIR [20] | 10.1 | 19.1 | 28.1 | 7.9 | 16.3 | 25.7 | 17.9 |
| CIG [60] | 10.6 | 19.2 | 27.4 | 7.9 | 16.9 | 25.4 | 17.9 |
| **CLIP+CRL** | 15.4 | 26.7 | 35.8 | 17.0 | 27.8 | 37.8 | 26.8 |
| Combiner* [55] | 16.6 | 27.7 | 37.2 | 18.0 | 32.2 | 41.6 | 28.9 |
| **CLIP+CRL***  | **19.7** | **32.7** | **41.3** | **21.0** | **35.9** | **44.8** | **32.6** |

**Baseline.** We first compare CRL with two traditional clustering methods, CC [32] and SCAN [54]. Furthermore, we incorporate Multi-Map [68], a customized clustering approach that leverages the CLIP model, into the comparison. Additionally, we report the performance of k-means clustering applied to the image representation of CLIP, ALIGN and MetaCLIP, to provide an intuitive baseline analysis.

**Performance.** As shown in Table. 2, CRL gains consistent performance improvement compared with the original CLIP, ALIGN and MetaCLIP. In particular, CRL obtains an ACC boost of CLIP over 75% on the color criterion. This improvement can be better visualized by T-SNE [53], as shown in the Appendix. Though traditional clustering methods exhibit some superiority on the "shape" criterion, CRL achieves consistently better results on other criteria. This implies that traditional clustering methods have a strong bias towards a single criterion, yet lack the flexibility and capability to cluster data based on other meaningful criteria.

## 4.2 Customized Retrieval

For customized retrieval, we also conduct experiments on its two subtasks, namely, customized similarity retrieval and customized fashion retrieval. Given a query image and a condition object, customized similarity retrieval aims to retrieve the most conditionally similar image from candidates, as illustrated in Fig. 4. As shown in Fig. 5, customized fashion retrieval searches all candidate images of fashion items, which share the same value as the query image under the specific criterion.

### 4.2.1 Customized Similarity Retrieval

**Dataset.** We adopt GeneCIS[55] as the benchmark for this task, which comprises two settings. As shown in Fig. 4, (a) "Focus" setting aims to retrieve the candidate that contains both the same scene (*e.g.*, living room) and the condition object (*e.g.*, table) as the query image. (b) In contrast, the "Change" setting requires the target image to maintain the same scene (*e.g.*, railway) as the query image while including the condition object (*e.g.*, tree) that is absent in the query image.

**Setup.** This benchmark involves two factors, namely, object (text condition) and scene (query image). To employ CRL, we ask the LLM for the common scenes, obtaining the conditional representation of the query and candidate images. Then we calculate and sum the similarities of these two factors for retrieval. This operation is detailed in the Appendix. Additionally, we use ViT-B/16 pre-trained on CLIP as the backbone, following the previous work [55].

**Metric.** The recall rates R@1, R@2, and R@3 serve as the evaluation metrics for this task. Higher recall rates imply better retrieval results.

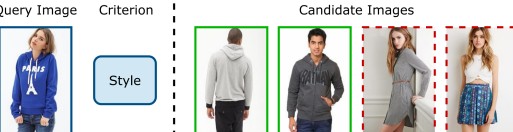

Figure 5: An example of the customized fashion retrieval task. Given a criterion, it searches all the candidate images that share the same value as the query image.

Table 4: Performance on the task of customized fashion retrieval. The symbol $^\dagger$ signifies that no training is conducted.

| Method | Texture | Fabric | Shape | Part | Style | Mean |
|---|---|---|---|---|---|---|
| Random | 6.69 | 2.69 | 3.23 | 2.55 | 1.97 | 3.38 |
| Triplet [57] | 13.26 | 6.28 | 9.49 | 4.43 | 3.33 | 7.36 |
| CSN [57] | 14.09 | 6.39 | 11.07 | 5.13 | 3.49 | 8.01 |
| ASEN [39] | 15.13 | 7.11 | 12.39 | 5.51 | 3.56 | 8.74 |
| ASEN++ [11] | 15.60 | 7.67 | 14.31 | 6.60 | 4.07 | 9.64 |
| RPF [12] | 15.62 | 8.30 | 15.02 | 7.38 | 4.77 | 10.22 |
| CLIP [48] | 9.14 | 4.68 | 7.86 | 4.26 | 4.48 | 6.08 |
| **CLIP+CRL**$^\dagger$ | 11.03 | 6.76 | 11.80 | 5.56 | 4.42 | 7.93 |
| **CLIP+CRL** | **16.88** | **9.31** | **16.98** | **7.54** | **5.95** | **11.33** |

**Baseline.** Following [55], we first provide three simple CLIP-only baselines, namely CLIP$_{\text{image}}$, CLIP$_{\text{text}}$ and CLIP$_{\text{image+text}}$, detailed in the Appendix. In addition, we include five retrieval baselines Pic2Word [49], SEARLE [4], LinCIR [20], CIG [60] and Combiner [55] for benchmarking. Notably, Combiner leverages the external dataset CC3M [51] to fine-tune the CLIP model. Thus we evaluate the performance of CRL under two scenarios: using the original CLIP weights and using the weights fine-tuned by Combiner.

**Performance.** As can be observed from Table. 3, CRL demonstrates substantial improvements over the original CLIP baselines, achieving a notable gain of $6.9\%$ in the mean recall. When leveraging fine-tuned CLIP weights, CRL further extends its advantage, surpassing Combiner by $3.7\%$ in mean recall, simultaneously maintaining consistent performance gains across all metrics.

#### 4.2.2 Customized Fashion Retrieval

**Dataset.** Following previous works [39], we use the category and attribute prediction benchmark of DeepFashion [36] as the evaluation dataset for this task, which consists of 221k / 27k / 27k images for training / validating / testing. This benchmark has 5 criteria, namely, "texture", "fabric", "shape", "part" and "style", with $156, 218, 180, 216,$ and $230$ values, detailed in the Appendix.

**Setup.** We first obtain the embeddings by CRL in a training-free manner. After that, we seamlessly append a two-layer MLP to the embeddings, subsequently training this MLP and the backbone. The training process is detailed in the Appendix. In addition, following previous works, ViT-B/16 is adopted as the backbone for this task.

**Metric.** Following existing works, we use the Mean Average Precision (MAP) as the evaluation metric for the customized fashion retrieval task. Higher MAP values indicate better retrieval results.

**Baseline.** We first add a Random baseline, which randomly sorts all the candidate images. Moreover, we provide a Triplet baseline, which uses the standard triplet ranking loss [57] to train a joint embedding space. Further, we compare CRL with $5$ state-of-the-art fashion retrieval methods, including Triplet [57], CSN [57], ASEN [39], ASEN++ [11] and RPF [12]. Besides, we also provide a CLIP baseline that embeds all the images with the image encoder.

**Performance.** As shown in Table 4, CRL achieves notable improvements over the CLIP baseline in a training-free manner, with a relative mean MAP gain of $30\%$.

Once the training is completed, CRL establishes new state-of-the-art performance, surpassing the best competitive method RPF by $10\%$ relativelty in mean MAP. These results further validate CRL's effectiveness in customized tasks and its compatibility with model fine-tuning strategies.

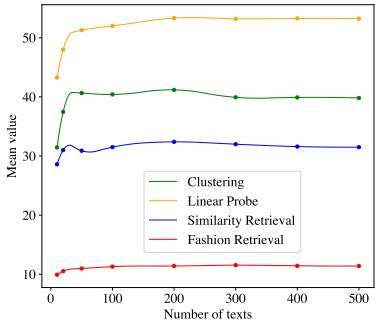

Figure 6: Performance with different numbers of texts.

### 4.3 Analysis on Textual Basis

To prove the robustness of CRL, we examine CRL's performance based on the CLIP model for the above-mentioned four customized tasks under varying levels of textual basis. To be specific, we explicitly control the number of generated descriptive texts and report the mean value of each task here, while the complete results can be seen in the Appendix. As Fig. 6 shows, CRL achieves stable performance for different numbers of texts except when the number is too small. In other words, CRL is a robust method for various tasks, as long as there is a reasonable number of descriptive texts to establish the semantical basis for the customized space. More ablation studies can be found in the Appendix.

## 5 Limitation

Based on our observations and experiments, we found that our method suffers from two main limitations. Firstly, despite its generalizability across different criteria, it may not outperform clustering methods like CC and SCAN under the universal criterion "shape". This is likely because these methods employ specially targeted designs for clustering under this criterion. Anyway, we acknowledge that CRL is not optimal on the universal criterion. Secondly, our method only roughly approximates the basis. We've tried various strategies to filter the texts generated by the LLM, but none have proven to be effective across all criteria. Nevertheless, we are confident that better strategies could be devised to acquire the text basis.

## 6 Conclusion

In this paper, we identify a fundamental limitation of existing representation learning methods: they predominantly derive universal embeddings that capture the most salient semantic features, making them suboptimal for customized tasks that prioritize non-dominant semantics. To address this, we propose CRL, a simple yet effective conditional representation learning method that adapts the universal representation to specific criteria through a basis transformation process. In brief, CRL utilizes a large language model (LLM) and a vision-language model (VLM) to generate textual descriptors that are semantically aligned with the user-specified criterion. These descriptors form an interpretable text basis, guiding the transformation of the image representation to enhance its expressiveness under the given criterion. Extensive experiments validate the effectiveness and generality of CRL across diverse tasks and criteria. By shifting the focus toward conditional representation learning, an underexplored yet promising paradigm, we hope this work could spark new insights and foster further research in this direction.

## Acknowledgements

This work was supported in part by NSFC under Grant 62176171, U21B2040, 623B2075, 62472295; in part by China National Postdoctoral Program for Innovative Talents under Grant BX20250392; in part by the Fundamental Research Funds for the Central Universities under Grant CJ202303; and in part by Sichuan Science and Technology Planning Project under Grant 24NSFTD0130.

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

# Appendix

In the Appendix, we provide supplementary information on the four customized tasks that are briefly introduced in the main paper. The Appendix is organized according to these four tasks, with each section dedicated to elaborating on one specific task. In the end, we provide ablation experiments for different LLMs, temperatures, and prompts of the LLM and VLM.

## A    Customized Few-shot Learning

### A.1    Dataset Description

We adopt Clevr-4 [56] and Cards [67] as benchmark datasets for this task. Based on the CLEVR dataset [26], Clevr-4 is a synthetic benchmark that introduces four distinct yet equally valid groupings of the data, namely, "texture", "shape", "color" and "count". It employs computer graphics tools to generate images featuring multiple objects positioned within fixed scenes, as shown in Fig. 7. As for Cards, it contains 8,029 images of poker cards, categorized along two independent dimensions: card number (such as Ace, King, Queen) and suit type (clubs, diamonds, hearts, spades).

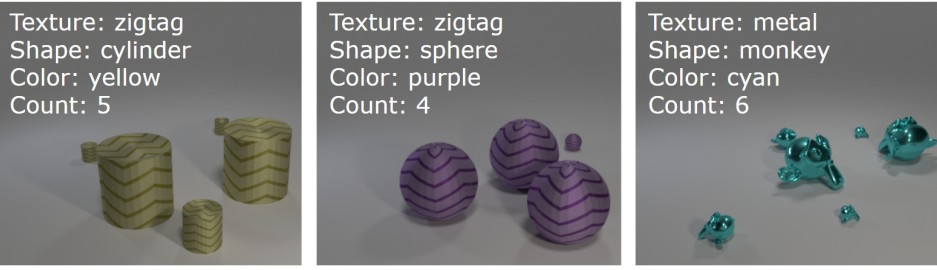

Figure 7: Three examples of clevr4 dataset.

### A.2    Prompt and Response

For the task of customized few-shot learning, we take the criterion of "Color" as an example.

### A.2.1    LLM Prompt

```
Please generate common expressions to describe the color, as many as
possible, formatted as:  ["...", "...", "..."].  Ensure all items
are unique and written in a single line, without any nested lists or
additional formatting.  You may describe the same color in different ways,
such as "red", "crimson", or "scarlet".  Only generate the list, and do
not include any additional information.
```

where the underlined part corresponds to some example synonyms to encourage diversity in the generated descriptive texts.

In addition, to control the number of the generated texts by the GPT-4, we modify the prompt as follows:

```
Please generate 100 expressions to describe the color, formatted as:
["...", "...", "..."].  Ensure all items are unique and written in a
single line, without any nested lists or additional formatting.  You may
describe the same color in different ways, such as "red", "crimson", or
"scarlet".  Only generate the list, and do not include any additional
information.
```

Due to the output length limitation of the LLM, we need to repeat the inquiry several times until obtaining the specific number of generated texts. Since this modification is very slight, we only present it here once for clarity.

### A.2.2 LLM Response

```
["red", "crimson", "scarlet", "ruby", "cherry", "rose", "burgundy", "cardinal", "wine",
"firetruck red", "ferrari red", "carmine", "sangria", "apple red", "poppy", "raspberry",
"pink", "blush", "fuchsia", "magenta", "hot pink", "baby pink", "coral", "peach", "apricot",
"salmon", "light pink", "powder pink", "lavender", "violet", "indigo", "purple", "amethyst",
"grape", "orchid", "eggplant", "plum", "lilac", "mauve", "periwinkle", "blue", "navy", "azure",
"sky blue", "cobalt", "cerulean", "sapphire", "electric blue", "royal blue", "baby blue",
"powder blue", "teal", "turquoise", "aqua", "seafoam", "green", "emerald", "forest green",
"mint", "lime", "olive", "kelly green", "sage", "pea green", "jade", "chartreuse", "yellow",
"gold", "lemon", "butter", "canary", "mustard", "sunflower", "cream", "ivory", "beige", "tan",
"coffee", "chocolate", "brown", "copper", "rust", "mahogany", "walnut", "espresso", "gray",
"slate", "charcoal", "pewter", "silver", "platinum", "black", "coal", "ebony", "jet black",
"onyx", "snow", "ivory", "white", "pearl", "bone", "eggshell", "vanilla"]
```

### A.2.3 Ground Truth Label

```
["gray", "red", "blue", "green", "brown", "purple", "cyan", "yellow", "pink", "orange"]
```

### A.2.4 VLM Prompt

```
        Objects with the color of red.
        Objects with the color of green.
        Objects with the color of blue.
                        ......
```

## A.3 Experimental Setting

After multiplying the same text basis, the gap between images shrinks. To accelerate the few-shot learning process, we normalize the transformed conditional representation to have zero mean and unit variance.

## A.4 Performance

As shown in Table 5, CRL achieves stable few-shot learning results under different numbers of LLM-generated descriptive texts, except when the text number is too small.

# B Customized Clustering

## B.1 Dataset, Prompt and Response

For the task of customized clustering, we use the same datasets, prompts and LLM responses as the customized few-shot learning task. Therefore, we omit the repeated descriptions here.

## B.2 Improvement Visualization

CRL achieves the representation projection from the original feature space (which is often dominated by the "shape" criterion) to the conditional feature space, making it more expressive under the specified criterion. Fig. 8 shows the T-SNE visualizations of the original CLIP representation and CRL representation, from which one can clearly observe the improvement.

## B.3 Performance

CRL maintains consistent clustering performance under different quantities of LLM-generated texts, as presented in Table 6, with a drop only when the number of texts is very limited.

Table 5: Customized few-shot learning performance under different numbers of texts.

| Text-num | Clevr4-10k | | | | | | | | | Mean |
|---|---|---|---|---|---|---|---|---|---|---|
| | Texture | | | Shape | | | Color | | | |
| | 1 | 5 | 10 | 1 | 5 | 10 | 1 | 5 | 10 | |
| 10 | 16.98 | 25.79 | 29.68 | 52.86 | 71.87 | 78.59 | 47.09 | 70.19 | 75.75 | 52.09 |
| 20 | 18.02 | 30.04 | 36.25 | 53.51 | 75.78 | 82.79 | 55.02 | 83.05 | 88.02 | 58.05 |
| 50 | 18.58 | 34.67 | 44.01 | 56.82 | 82.19 | 88.57 | 61.81 | 86.70 | 91.57 | 62.77 |
| 100 | 19.19 | 36.73 | 47.11 | **57.47** | 84.64 | 91.13 | 61.86 | 87.12 | 92.20 | 64.16 |
| 200 | 20.49 | 39.23 | 50.08 | 54.96 | 84.38 | 91.48 | **66.82** | **89.60** | **93.68** | **65.64** |
| 300 | 20.37 | 39.64 | 50.82 | 55.04 | 84.62 | 91.34 | 65.80 | 88.90 | 93.28 | 65.53 |
| 400 | **20.54** | **39.94** | **51.14** | 55.39 | **84.95** | 91.57 | 65.16 | 88.51 | 93.07 | 65.59 |
| 500 | 20.36 | 39.83 | 50.98 | 55.09 | 84.86 | **91.59** | 64.44 | 88.11 | 92.83 | 65.34 |

| Text-num | Clevr4-10k | | | Cards | | | | | | Mean |
|---|---|---|---|---|---|---|---|---|---|---|
| | Count | | | Number | | | Suits | | | |
| | 1 | 5 | 10 | 1 | 5 | 10 | 1 | 5 | 10 | |
| 10 | **23.94** | **30.94** | **33.31** | 14.35 | 31.67 | 36.61 | 33.79 | 50.44 | 55.25 | 34.48 |
| 20 | 22.68 | 29.49 | 31.68 | 16.01 | 39.40 | 46.85 | 37.14 | 56.70 | 61.75 | 37.97 |
| 50 | 22.34 | 29.05 | 32.01 | 16.08 | 40.75 | 48.95 | 37.52 | 62.43 | 69.25 | 39.82 |
| 100 | 21.92 | 28.64 | 31.58 | 15.43 | 39.53 | 48.99 | **37.80** | 63.69 | 71.04 | 39.85 |
| 200 | 22.06 | 28.18 | 31.16 | 16.52 | 41.81 | 51.63 | 37.26 | **66.66** | **73.89** | 41.02 |
| 300 | 21.11 | 27.56 | 30.71 | 16.75 | 42.30 | 52.95 | 36.41 | 66.33 | 73.86 | 40.89 |
| 400 | 21.53 | 28.04 | 31.01 | 16.82 | 42.50 | 53.68 | 35.89 | 65.90 | 73.36 | 40.97 |
| 500 | 21.48 | 28.04 | 31.06 | **16.90** | **43.22** | **54.24** | 35.73 | 65.82 | 73.56 | **41.12** |

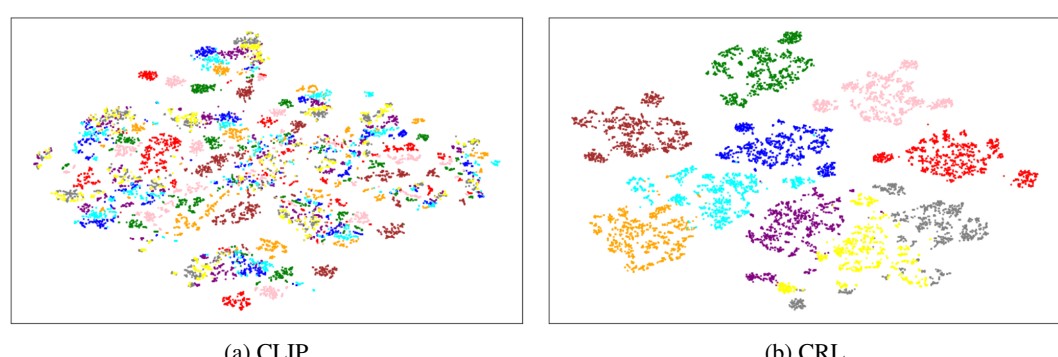

(a) CLIP      (b) CRL

Figure 8: T-SNE visualizations of the representations obtained by CLIP and CRL, under the "color" criterion of the Clevr4-10k dataset.

## C  Customized Similarity Retrieval

For this task, the criterion is "Scene." Below, we present both the prompt used and the corresponding results generated. It's worth noting that there are no ground truth labels for this task.

### C.1  Dataset Description

In both settings, the candidate images are required to share the same scene as the query image and satisfy the given object condition. The difference lies in the presence of the object condition in the query image. In the "Focus on an object" setting, the query image contains the object condition, while in the 'Change an object' setting, the query image doesn't. In other words, the "Focus" setting retrieves a positive target, while the "Change" setting searches for a negative target. Both settings consist of $1,960$ query images sourced from the classical CoCo [33] dataset, with each query image corresponding to 10-15 candidate images in the gallery.

Table 6: Customized clustering performance under different numbers of texts.

| Text-num | Texture | | | Shape | | | Color | | | Mean |
|---|---|---|---|---|---|---|---|---|---|---|
| | NMI | ACC | ARI | NMI | ACC | ARI | NMI | ACC | ARI | |
| | | | | | Clevr4-10k | | | | | |
| 10 | 12.79 | 26.37 | 7.32 | 67.27 | 66.68 | 54.59 | 55.50 | 58.23 | 40.50 | 43.25 |
| 20 | **13.57** | **26.40** | **8.04** | 67.92 | 68.86 | 57.00 | 76.71 | 78.74 | 67.79 | 51.67 |
| 50 | 11.94 | 24.25 | 6.76 | 74.89 | 78.64 | 67.08 | 85.90 | 86.11 | 78.88 | 57.16 |
| 100 | 10.62 | 23.29 | 5.88 | **77.72** | **80.61** | **70.41** | 85.20 | 82.92 | 76.27 | 56.99 |
| 200 | 13.16 | 26.49 | 7.97 | 75.71 | 78.78 | 67.34 | **88.73** | **86.68** | **81.40** | **58.47** |
| 300 | 12.58 | 25.46 | 7.39 | 74.78 | 76.28 | 66.18 | 88.06 | 86.40 | 80.58 | 57.52 |
| 400 | 11.90 | 24.91 | 7.12 | 74.79 | 78.58 | 67.19 | 87.92 | 85.07 | 80.33 | 57.53 |
| 500 | 11.13 | 24.44 | 6.64 | 74.13 | 77.62 | 66.58 | 87.66 | 86.61 | 80.51 | 57.26 |

| Text-num | Count | | | Number | | | Suits | | | Mean |
|---|---|---|---|---|---|---|---|---|---|---|
| | Clevr4-10k | | | | | Cards | | | | |
| | NMI | ACC | ARI | NMI | ACC | ARI | NMI | ACC | ARI | |
| 10 | **27.14** | **32.08** | **15.30** | **22.50** | **26.72** | **10.54** | 4.61 | 33.72 | 4.18 | 19.64 |
| 20 | 25.03 | 27.83 | 12.62 | 18.35 | 22.88 | 9.02 | 24.21 | 50.01 | 19.44 | 23.27 |
| 50 | 21.33 | 26.13 | 11.57 | 17.86 | 23.03 | 9.47 | 27.14 | 56.68 | 23.78 | **24.11** |
| 100 | 21.90 | 25.64 | 11.31 | 16.51 | 21.61 | 8.25 | 27.67 | **57.03** | 24.64 | 23.84 |
| 200 | 20.41 | 24.51 | 10.19 | 17.70 | 22.72 | 8.90 | **29.84** | 53.96 | **26.86** | 23.90 |
| 300 | 15.79 | 22.59 | 7.80 | 16.95 | 22.00 | 8.23 | 28.48 | 53.62 | 25.60 | 22.34 |
| 400 | 17.58 | 23.71 | 8.66 | 15.79 | 20.47 | 7.57 | 28.35 | 53.17 | 25.26 | 22.28 |
| 500 | 17.33 | 23.38 | 8.58 | 15.89 | 20.65 | 7.61 | 28.69 | 53.62 | 25.65 | 22.38 |

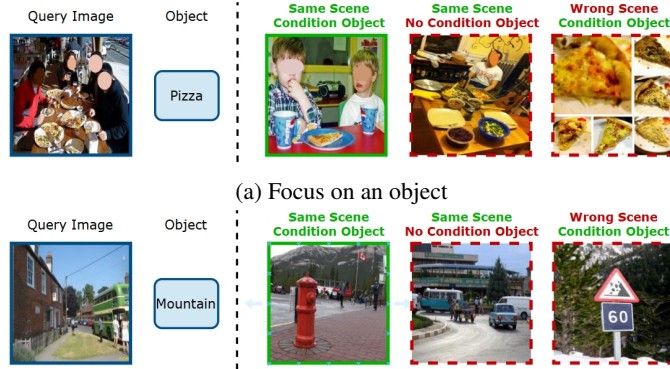

(a) Focus on an object

(b) Change an object

Figure 9: Two settings of the customized similarity retrieval task.

## C.2 Prompt and Response

### C.2.1 LLM Prompt

```
Please generate common expressions to describe the scene, as many as
possible, formatted as: ["...", "...", "...", "..."]. Ensure all
items are unique and written in a single line, without any nested
lists or additional formatting. You may describe the same scene in
different ways, such as "a cozy living room", "a snug lounge", or
"a warm and inviting sitting area". Only generate the list, and do not
include any additional information.
```

### C.2.2 LLM Response

```
["a spacious living room", "a cozy lounge", "a snug sitting area", "a well-lit kitchen", "a
modern dining room", "a stylish bedroom", "an airy home office", "a rustic cabin interior",
"a minimalistic studio apartment", "a vibrant indoor garden", "a home library", "a charming
balcony", "a serene patio", "a lush backyard", "a sunlit garden", "a peaceful park", "a scenic
mountain view", "a bustling city street", "a quiet suburban street", "a sunny beach", "a
```

serene lakeside view", "a dense forest", "a tranquil meadow", "a wild desert landscape", "a foggy morning in the woods", "a snowy winter scene", "a lively outdoor market", "a cozy backyard barbecue", "a sun-drenched terrace", "a secluded countryside lane", "a colorful street art scene", "a tropical rainforest", "a modern rooftop terrace", "an open-air café", "a bright sunny veranda", "a quiet riverbank", "a sunny playground", "a rugged coastline", "a picturesque vineyard", "a lively park", "a quiet pond", "a bustling urban square", "a scenic hillside view", "a charming street corner", "an expansive field of flowers", "a peaceful picnic spot", "a lively street festival", "a serene walking path", "a bustling town square", "a remote mountain pass", "a tranquil beachfront cabin", "a sunny picnic area", "a dense jungle trail", "a charming country road", "a misty hilltop view", "a windy coastal cliff", "a lush tropical beach", "an open farm field", "a quiet coastal town", "a vibrant city plaza", "a bustling shopping district", "a colorful flower garden", "a charming riverside walk", "a large public park", "a peaceful countryside landscape", "a warm outdoor terrace", "an outdoor wedding venue", "a lively amusement park", "a remote fishing village", "a scenic coastal highway", "a green city park", "a vibrant botanical garden", "a sandy dune landscape", "a cozy cabin porch", "a charming old town square", "a quiet beachside café", "a sunny urban terrace", "a tranquil garden courtyard", "a vibrant farmers' market", "a lively carnival scene", "a remote mountain village", "a rustic barnyard", "a calm rural road", "a dramatic cliffside view", "a lively beachfront promenade", "a peaceful lakeshore cabin", "a quiet rural pathway", "a lush botanical park", "a scenic hiking trail", "a lively town market", "a secluded desert oasis", "a colorful coastal town", "a quiet forest clearing", "a scenic boat dock", "a vast open field", "a serene cliffside walk", "a lively open-air concert", "a quiet hillside retreat", "a bright tropical beach", "a calm sandy shore", "a warm outdoor patio", "a charming outdoor café", "a vibrant city park", "a peaceful desert sunset", "a lush green terrace", "a rustic lakeside cabin", "a bright garden path", "a misty river valley", "a bustling port town", "a quiet mountain retreat", "a tranquil city courtyard", "a picturesque town harbor", "a lively street market", "a scenic desert plateau", "a quiet neighborhood street", "a charming seaside village", "a calm beachside retreat", "a dense evergreen forest", "a misty forest trail", "a sunny farm field", "a lively city park", "a charming cobblestone street", "a peaceful urban courtyard", "a tranquil village square", "a vibrant mountain town", "a lively marina", "a quiet waterfront view", "a peaceful countryside lane", "a cozy lakeside cabin", "a bright tropical garden", "a peaceful fishing spot", "a scenic valley view", "a lively park bench", "a quiet country lane", "a rustic vineyard", "a tropical outdoor pool", "a peaceful city park", "a lively outdoor music venue", "a calm lakeside dock", "a vibrant waterfront café", "a quiet garden path", "a sunny riverside park", "a scenic mountain road", "a charming forest cabin", "a tranquil rural vista", "a lively outdoor fair", "a calm and quiet pond", "a bustling pedestrian street", "a serene lakeside trail", "a lively street corner", "a bright and sunny lawn", "a rustic country garden", "a quaint village street", "a peaceful nature reserve", "a vibrant open-air market", "a scenic rural road", "a quiet forest retreat", "a sunny city rooftop", "a cozy open-air restaurant", "a secluded beach cove", "a lively harbor view", "a scenic lakeside trail", "a charming countryside path", "a lively town square", "a sunny country road", "a tranquil riverside path", "a busy urban park", "a rustic hillside cabin", "a scenic beach boardwalk", "a quiet rural farm", "a peaceful coastal village", "a lively urban park", "a tranquil mountain valley", "a vibrant street fair", "a charming oceanfront path", "a quiet street corner", "a lush tropical garden", "a scenic hilltop view", "a quiet lakeside retreat", "a busy shopping district", "a calm and quiet garden", "a lively mountain town square", "a peaceful coastal bluff", "a vibrant outdoor market square", "a quiet nature trail", "a scenic mountain cabin", "a sunny desert trail", "a peaceful urban garden", "a vibrant outdoor community center", "a calm lakeshore view", "a tranquil city park", "a quiet riverside retreat", "a bustling urban plaza", "a serene oceanfront view", "a quiet hilltop vista", "a lively carnival parade", "a vibrant beach festival", "a peaceful orchard", "a sunny green park", "a charming beach house", "a scenic ocean drive", "a peaceful rural countryside", "a vibrant plaza scene", "a lively downtown street", "a quiet city park bench", "a colorful street festival", "a tranquil nature spot", "a sunny village square", "a bustling beachside promenade", "a rustic waterfront cabin", "a busy shopping mall entrance", "a charming lakeside promenade", "a scenic cliffside", "a quiet street park", "a colorful beach scene", "a lively beach party", "a quiet garden café", "a calm sandy shore", "a vibrant rooftop garden", "a serene lakeside dock", "a peaceful open field", "a quiet scenic trail", "a lively street performer", "a rustic forest retreat", "a scenic

```
city skyline view", "a peaceful ocean retreat", "a lively town gathering", "a busy seaside
boardwalk", "a scenic countryside village"]
```

### C.2.3  VLM Prompt

> A photo with a **scene** of **a spacious living room.**
>
> A photo with a **scene** of **a cozy lounge.**
>
> A photo with a **scene** of **a snug sitting area.**
>
> ......

### C.3  Experimental Setting

This benchmark involves an object factor (text condition) and a scene factor (query image). For the object factor, we directly compute the similarity $S_1$ between the CLIP representations of the text condition and candidate images. For the scene factor, we first ask the LLM for the common scenes. Leveraging these scene texts as the text basis, we can obtain all images' conditional representations by Eq. (4) in the main paper. Then, we calculate the similarity $S_2$ between the conditional representations of the query image and the candidate images. Finally, we select the candidate with the maximum combined similarity value $S = S_1 + \alpha * S_2$, where the weighting parameter $\alpha$ is set to 10.

### C.4  Baseline

$\text{CLIP}_\text{image}$ baseline employs the CLIP image encoder to generate embeddings for both query and gallery images, subsequently retrieving the most similar gallery image to the query. $\text{CLIP}_\text{text}$ adopts a cross-modal approach, where the textual condition is encoded by the CLIP text encoder while gallery images are processed through the image encoder, enabling retrieval based on text-image alignment. $\text{CLIP}_\text{image+text}$ computes the average of query image embeddings and condition text embeddings, which is then used for retrieval from the gallery space.

### C.5  Performance

Table 7 demonstrates that CRL performs robustly across a wide range of descriptive text quantities, with performance degradation observed only when the number of texts is insufficient.

Table 7: Customized similarity retrieval performance under different numbers of texts.

| Text-num | Focus | | | Change | | | Mean |
|---|---|---|---|---|---|---|---|
| | R@1 | R@2 | R@3 | R@1 | R@2 | R@3 | |
| 10 | 17.7 | 29.1 | 38.0 | 18.7 | 29.9 | 38.5 | 28.6 |
| 20 | 18.8 | 32.0 | 40.4 | 20.4 | 32.5 | 41.8 | 31.0 |
| 50 | 18.4 | 31.7 | 40.1 | 20.1 | 33.2 | 42.2 | 30.9 |
| 100 | 18.5 | 31.0 | **41.6** | 20.7 | 34.1 | 43.4 | 31.5 |
| 200 | **20.1** | **33.0** | 41.2 | **21.4** | **35.1** | **43.8** | **32.4** |
| 300 | 19.2 | 32.6 | 40.9 | 21.3 | 34.5 | 43.4 | 32.0 |
| 400 | 19.4 | 31.9 | 40.4 | 20.8 | 34.1 | 43.1 | 31.6 |
| 500 | 19.0 | 31.7 | 41.0 | 20.4 | 33.7 | 43.3 | 31.5 |

## D  Customized Fashion Retrieval

For the task of customized fashion retrieval, we take the criterion of "Texture" as an example. We provide the prompts, responses and ground truth labels.

### D.1  Dataset Description

DeepFashion [36] is a large-scale clothing dataset that provides four benchmarks, each tailored to a specific task. Following previous work [39], we use the category and attribute prediction split as the

benchmark. We provide a summary of the criteria for this dataset in Table. 8, listing some examples for each criterion. As shown in Fig. 10, given the criterion "style", this benchmark requires retrieving the candidate that shares the same value ("Mickey") as the query image.

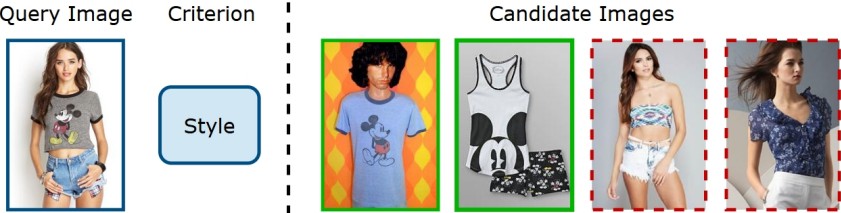

Figure 10: An example of the customized fashion retrieval task.

Table 8: A summary of the criteria of the customized fashion retrieval benchmark.

| Criterion | Value | Count |
|---|---|---|
| Texture | abstract, animal, baroque, camo, circle, ... | 156 |
| Fabric | acid, bead, canvas, cloud, feather, ... | 218 |
| Shape | a-line, boxy, cropped, flared, shirt, ... | 180 |
| Part | bell, bow, buttoned, cowl, flat, ... | 216 |
| Style | baseball, bike, eagle, fox, mickey, ... | 230 |

## D.2 Prompt and Response

### D.2.1 LLM Prompt

```
Please generate common expressions to describe the texture, as many
as possible, formatted as: ["...", "...", "..."]. Ensure all items
are unique and written in a single line, without any nested lists or
additional formatting. You may describe the same texture in different
ways, such as "baroque", "ornate", or "luxurious". Only generate the
list, and do not include any additional information.
```

### D.2.2 LLM Response

```
["smooth", "rough", "silky", "velvety", "grainy", "bumpy", "fuzzy", "soft", "coarse",
"textured", "polished", "matte", "glossy", "shiny", "satin-like", "woven", "embellished",
"embroidered", "fluffy", "sheer", "lacey", "ribbed", "pleated", "crinkled", "frosted",
"distressed", "weathered", "sparkling", "glittery", "glimmering", "luxurious", "elegant",
"ornate", "ornamental", "baroque", "rustic", "vintage", "gritty", "delicate", "intricate",
"smooth-to-the-touch", "textured-to-perfection", "matte-finish", "high-gloss", "fuzzy-finish",
"heavy-duty", "soft-touch", "pebbled", "canvas-like", "embroidered", "fringed", "knitted",
"crocheted", "tight-knit", "loose-knit", "structured", "fluid", "cloud-like", "slick",
"furry", "cozy", "snug", "plush", "velvety-smooth", "sandpaper-like", "suede", "nubuck",
"grippy", "twilled", "crinkled", "slubbed", "grainy-texture", "soft-grip", "scuffed",
"weathered-leather", "textured-leather", "crinkly", "pleated-finish", "waterproof",
"thick-threaded", "gossamer", "translucent", "woven-texture", "frayed", "tightly-woven",
"loose-woven", "threadbare", "matted", "dense-weave", "open-weave", "honeycomb", "cut-out",
"quilted", "pleated-texture", "smooth-leather", "grain-leather", "burnished"]
```

### D.2.3 Ground Truth Label

```
['abstract', 'abstract chevron', 'abstract chevron print', 'abstract diamond', 'abstract
floral', 'abstract floral print', 'abstract geo', 'abstract geo print', 'abstract paisley',
'abstract pattern', 'abstract print', 'abstract printed', 'abstract stripe', 'animal',
```

```
['animal print', 'bandana', 'bandana print', 'baroque', 'baroque print', 'bird', 'bird print',
'botanical', 'botanical print', 'boxy striped', 'breton', 'breton stripe', 'brushstroke',
'brushstroke print', 'butterfly', 'butterfly print', 'camo', 'camouflage', 'checked',
'checkered', 'cheetah', 'chevron', 'chevron print', 'chiffon floral', 'circle', 'clashist',
'classic striped', 'colorblock', 'colorblocked', 'crochet floral', 'daisy', 'daisy print',
'diamond', 'diamond print', 'ditsy', 'ditsy floral', 'ditsy floral print', 'dot', 'dots',
'dotted', 'embroidered floral', 'floral', 'floral flutter', 'floral paisley', 'floral pattern',
'floral print', 'floral textured', 'floral-embroidered', 'flower', 'foil', 'folk', 'folk
print', 'geo', 'geo pattern', 'geo print', 'geo stripe', 'giraffe', 'giraffe print', 'graphic',
'grid', 'grid print', 'heart', 'heart print', 'heathered stripe', 'houndstooth', 'ikat', 'ikat
print', 'kaleidoscope', 'kaleidoscope print', 'knit stripe', 'knit striped', 'leaf', 'leaf
print', 'leave', 'leopard', 'leopard print', 'linen', 'linen-blend', 'mandala', 'mandala
print', 'marble', 'marble print', 'marled', 'marled stripe', 'medallion', 'medallion print',
'mixed', 'mixed print', 'mixed stripe', 'mosaic', 'mosaic print', 'multi-stripe', 'nautical',
'nautical stripe', 'nautical striped', 'ombre', 'ornate', 'ornate paisley', 'ornate print',
'paint', 'paint splatter', 'painted', 'paisley', 'paisley print', 'palm', 'palm print',
'palm springs', 'palm tree', 'pattern', 'patterned', 'pinstripe', 'pinstriped', 'polka dot',
'pom-pom', 'print', 'print shirt', 'print woven', 'printed', 'ribbed stripe', 'ringer',
'rugby stripe', 'rugby striped', 'sophisticated', 'southwestern', 'southwestern-inspired',
'southwestern-patterned', 'southwestern-print', 'speckled', 'splatter', 'spotted', 'stripe',
'striped', 'stripes', 'structured', 'tonal', 'tribal', 'tribal-inspired', 'two-tone',
'varsity-striped', 'watercolor', 'zig', 'zigzag']
```

### D.2.4 VLM Prompt

```
A fashion with a texture of smooth.

A fashion with a texture of rough.

A fashion with a texture of silky.

          ......
```

### D.3 Experimental Setting

Following previous works, we exploit the triplet ranking loss to train this MLP and the backbone by 100k triplets, which are derived from the training split of the DeepFashion dataset. The training process consists of two stages. In the first stage, we only train the MLP and freeze the CLIP model for 1000 epochs, with an initial learning rate of 1e-4. In the second stage, we freeze the MLP and slightly fine-tune the CLIP model for 100 epochs, with a smaller initial learning rate of 1e-6. The optimizer, the decaying rate, the decaying step size and the triplet margin are set to Adam, 0.9, 3 and 0.3, respectively.

### D.4 Performance

As illustrated in Table 9, CRL exhibits consistent fashion retrieval performance across varying numbers of LLM-generated descriptive texts, except in cases where the number of texts is too small.

Table 9: Customized fashion retrieval performance under different numbers of texts.

| Text-num | Texture | Fabric | Shape | Part | Style | Mean |
|---|---|---|---|---|---|---|
| 10 | 15.80 | 8.15 | 14.40 | 6.49 | 4.80 | 9.93 |
| 20 | 16.21 | 8.93 | 15.18 | 7.07 | 5.06 | 10.52 |
| 50 | 16.64 | 9.02 | 16.48 | 7.10 | 5.66 | 10.98 |
| 100 | 17.01 | 9.24 | 16.58 | 7.55 | **6.17** | 11.30 |
| 200 | 17.14 | 9.25 | 17.20 | 7.42 | 6.05 | 11.40 |
| 300 | **17.28** | **9.42** | **17.32** | 7.64 | 6.10 | **11.54** |
| 400 | 17.05 | 9.38 | 17.06 | 7.58 | 6.07 | 11.42 |
| 500 | 16.68 | 9.40 | 17.14 | **7.67** | 6.13 | 11.40 |

# E Ablation Studies

## E.1 LLM

As for the selection of LLMs, we use the same prompt to query four mainstream LLMs: GPT-4o, Deepseek-v3, Gemini 2.5, and Claude 4. As can be seen in Table 10, our method does not particularly rely on any specific LLM.

Table 10: Customized classification performance under different LLMs.

| Task | GPT-4o [1] | Deepseek-v3 [34] | Gemini 2.5 [8] | Claude 4 [3] | Std |
|---|---|---|---|---|---|
| Clustering | 44.96 ± 0.52 | 43.40 ± 0.50 | 43.80 ± 0.55 | 43.75 ± 0.58 | 0.59 |
| Few-shot Learning | 53.34 ± 0.44 | 52.94 ± 0.40 | 52.93 ± 0.41 | 53.43 ± 0.45 | 0.23 |

## E.2 Temperature

As for the LLM temperature $t$, we set $t$ to $0, 0.5, 1, 1.5$ to obtain the text basis, respectively. The temperature ranges from $0$ to $2$, with higher values introducing more variability and randomness in the LLM's output. When the temperature approaches $2$, the generated content becomes almost entirely random, so we did not include this setting in our experiments. The experimental results in Table 11 validate the robustness of our method to the temperature parameter.

Table 11: Customized classification performance under different temperatures.

| Task | t=0 | t=0.5 | t=1 | t=1.5 | Std |
|---|---|---|---|---|---|
| Clustering | 43.33 ± 0.73 | 43.74 ± 0.66 | 44.96 ± 0.52 | 43.27 ± 0.39 | 0.68 |
| Few-shot Learning | 53.07 ± 0.49 | 53.27 ± 0.48 | 53.34 ± 0.44 | 52.50 ± 0.41 | 0.33 |

## E.3 LLM Prompt

As for the LLM prompt, we require it to include the [criterion]. We devise below 5 different templates:

1) Generate common expressions to describe the [criterion].
2) List a wide variety of typical phrases used to characterize the [criterion].
3) Enumerate familiar terms or expressions people often use when referring to the [criterion].
4) Identify and list expressions frequently used to convey the concept of the [criterion].
5) How do people usually talk about the [criterion]?

One can observe from Table 12 that different LLM prompts can yield close performance improvements, indicating that our method is robust against the LLM prompt.

Table 12: Customized classification performance under different LLM prompts.

| Task | Prompt 1 | Prompt 2 | Prompt 3 | Prompt 4 | Prompt 5 | Std |
|---|---|---|---|---|---|---|
| Clustering | 44.96 ± 0.52 | 42.45 ± 0.44 | 42.87 ± 0.31 | 44.75 ± 0.65 | 42.60 ± 0.55 | 1.10 |
| Few-shot Learning | 53.34 ± 0.44 | 52.95 ± 0.39 | 53.42 ± 0.37 | 52.82 ± 0.48 | 52.40 ± 0.44 | 0.37 |

## E.4 VLM Prompt

As for the VLM prompt, we require it to contain the [criterion] and the generated [text] by the LLM. We also devise below 5 different templates:

1) objects with the [criterion] of [text]
2) a photo with the [criterion] of [text]

3) itap with the [criterion] of [text]

4) art with the [criterion] of [text]

5) a cartoon with the [criterion] of [text]

As suggested in the Table 13, our method remains stable across different VLM prompts.

Table 13: Customized classification performance under different VLM prompts.

| Task | Prompt 1 | Prompt 2 | Prompt 3 | Prompt 4 | Prompt 5 | Std |
|---|---|---|---|---|---|---|
| Clustering | 44.96 ± 0.52 | 43.72 ± 0.44 | 44.78 ± 0.46 | 43.07 ± 0.53 | 42.33 ± 0.49 | 1.00 |
| Few-shot Learning | 53.34 ± 0.44 | 52.28 ± 0.38 | 52.56 ± 0.42 | 53.48 ± 0.38 | 52.58 ± 0.23 | 0.47 |

