# OpenReview forum: "Conditional Representation Learning for Customized Tasks"
_NeurIPS.cc/2025/Conference — NeurIPS 2025 spotlight_

### Official Review · Reviewer_GS6R · 2025-06-29

**Clarity:** 3
**Significance:** 3
**Originality:** 3
**Rating:** 5
**Confidence:** 5

**Summary:**

This work proposes a conditional representation learning method, which transforms the universal image embeddings into customized ones catering to the criteria of downstream tasks. Experiments on both supervised/unsupervised classification and retrieval demonstrate the effectiveness and superiority of the proposed method.

**Questions:**

I expect the authors to address my concerns in the weaknesses section. Besides, what are the differences between the task of "Customized Similarity Retrieval" in the paper and the task of "compositional retrieval"?

**Ethical Concerns:**

["NO or VERY MINOR ethics concerns only"]

**Limitations:**

Yes.

**Quality:**

3

**Strengths And Weaknesses:**

Strengths
1. The proposed concept of conditional representation learning is very interesting. As the authors pointed out, most existing representation learning methods learn universal embeddings, which could be suboptimal for customized tasks that prioritize non-dominant semantics. Instead of resorting to laborious fine-tuning, this work reveals that a simple but ingenious basis transformation operation could uncover semantics with respect to a specific criterion.
2. The proposed method is intuitively and technically correct. Its simple but effective design makes it a plug-and-play module for various customized downstream tasks, as validated in the experiments.
3. The paper is well written in general, making it easy to follow the key idea.

Weaknesses
1. Currently, the nouns used for constructing the textual basis are simply generated based on the given criteria. However, intuitively, the best set of textual basis could differ for different input images, even under the same criterion. I encourage the authors to take that into consideration and at least discuss such a potential limitation of this method.
2. I would like to see the quality of the generated texts. Since the prompt is simply "generate as many as possible", would the generated words be replicated and redundant? How does that influence the final performance? For example, I expect experimental results on a de-duplicated text basis.
3. For the current classification evaluation, it is more accurate to highlight "few-shot learning" instead of "supervised linear probing".

---

> ### Author Rebuttal · Authors · 2025-07-31
>
> Thanks for your recognition of the problem this work aims to address, as well as our simple yet effective idea. We sincerely appreciate your positive feedback. Below are the point-by-point responses to your concerns mentioned in the weaknesses and questions sections.
>
> **Weakness 1: Best set of textual basis**
>
> We appreciate the insightful comment. We will try to adopt a unified selection strategy in this work (e.g., variance, distance, PCA, maximal connected subgraph, etc.). We hold the same view that this is a valuable direction, and we will consider it in future explorations.
>
> **Weakness 2: Quality of the generated texts**
>
> Our method only requires that the generated texts roughly describe the given criterion, without the need for precise or exhaustive phrasing.
>
> Take the "color" criterion as an example, the generated texts are:
>
> ["red", "crimson", "scarlet", "ruby", "cherry", "rose", "burgundy", "cardinal", "wine", "firetruck red", "ferrari red", "carmine", "sangria", "apple red", "poppy", "raspberry",  "red", "pink", "blush", "fuchsia", "magenta", "hot pink", "baby pink",  "pink", "coral", "peach", "apricot", "salmon", "light pink", "powder pink", "lavender", "violet", "indigo", "purple", "amethyst", "grape", "orchid", "eggplant", "plum", "lilac", "mauve", "periwinkle", "blue", "navy", "azure", "sky blue", "cobalt", "cerulean", "sapphire", "electric blue", "royal blue", "baby blue", "powder blue", "blue", "teal", "turquoise", "aqua", "seafoam", "green", "emerald", "forest green", "mint", "lime", "olive", "kelly green", "sage", "pea green", "jade", "chartreuse", "green", "yellow", "gold", "lemon", "butter", "canary", "mustard", "sunflower", "cream", "ivory", "beige", "tan", "coffee", "chocolate", "yellow", "brown", "copper", "rust", "mahogany", "walnut", "espresso", "gray", "slate", "charcoal", "pewter", "silver", "platinum", "black", "coal", "ebony", "jet black", "onyx", "snow", "ivory", "white", "pearl", "bone", "eggshell", "vanilla"]
>
> Some generated texts are duplicated ("red", "blue", etc.), and we'll filter out these redundant texts. As the table below shows, this brings a slight improvement in clustering performance.
>
> | Metric | Original Texts | Texts without replication |
> | :----: | :------------: | :-----------------------: |
> |  NMI   |     88.18      |           88.67           |
> |  ACC   |     87.42      |           88.05           |
> |  ARI   |     81.27      |           82.30           |
>
> As for the redundancy, we explore the number of generated words in Figure 6 in the main text. As depicted, our method achieves stable performance, demonstrating its robustness to the redundancy of the generated words.
>
> **Weakness 3: More appropriate title**
>
> Thank you for pointing this out — we agree that framing the classification evaluation as “few-shot learning” is more precise than “supervised linear probing”. We will modify it in the next version.
>
> **Question 1: Comparison to compositional retrieval**
>
> The customized similarity retrieval task is more closely associated with directed representation transformation, where the criterion is fixed.
>
> In comparison, compositional retrieval does not involve such a targeted transformation or toward a specific criterion. Instead, it typically involves retrieving images from the gallery based on the combination of multiple texts and the given image.
>
> In future work, we also plan to explore how our method can be extended or adapted to compositional retrieval scenarios.
>
>
>
> We will add the above discussions in the next version.
>
> Finally, we sincerely thank you for your positive feedback and insightful comments again, which are encouraging and helpful in guiding the future development of our work.

---

> > ### Comment · Reviewer_GS6R · 2025-08-07
> > **Thanks to the author's response**
> >
> > Thanks to the author's response, which addressed my concerns about the quality of the generated text and the definition of classification in the validation experiments. Therefore, I will maintain my score.

---

> > > ### Author Response · Authors · 2025-08-07
> > >
> > > Thanks again for your positive feedback and recognition of our work. We sincerely appreciate your support and will improve our paper following your advice.

---

### Official Review · Reviewer_YAxP · 2025-07-02

**Clarity:** 3
**Significance:** 3
**Originality:** 2
**Rating:** 4
**Confidence:** 3

**Summary:**

The authors introduce conditional representation learning, which aims to learn a representation conditioned on user-specified criteria. They frame this as a transformation of basis, using LLM-generated descriptors generated from the user's text input passed through a VLM as the basis. They pass the query images through the VLM and project those representations onto the text vector basis, essentially obtaining a vector of similarities to each of the text descriptors, which they use as their initial representation. They train MLPs on top of these. They suggest these representations are more suited to the downstream tasks related to the user's query.  They evaluate on various 'customized' variants of tasks such as classification, clustering, and retrieval.

**Questions:**

- This approach applies to methods like CLIP which produce image embeddings and text embeddings separately, with fusion only at the final embedding similarity. How does this compare to methods that create image representations conditioned on text already? What are the pros and cons?
- What differentiates the CRL task from previous related tasks?
- Are there any existing tasks that could be framed in this setting for broader comparison?
- With how many examples do the improvements level off?
- How robust is the approach to variations in the user-specified input? What is that input for the evaluated tasks?
- How does it compare to learned visual prompt approaches?

**Ethical Concerns:**

["NO or VERY MINOR ethics concerns only"]

**Final Justification:**

The authors provide a convincing justification for why this task customization setting is of interest, and answer many other of the original remaining questions, so I increase my score to borderline accept. Given the issues of task realism -- none of the evaluations are established and still seem a bit artificial to me -- I'm still not inclined higher than borderline and could see it going either way.

**Limitations:**

The authors say the limitations are in the experiments section, but I didn't catch it if so.

**Quality:**

3

**Strengths And Weaknesses:**

Strengths:
- The idea is simple to implement.
- It seems effective for the presented tasks.
- It provides a nice way to condition image representations on the text for the task with models where these representations are made separately.

Weaknesses:
- One of the stated main contributions is the conditional representation learning task, but there is a substantial body of work on task-conditioned and goal-conditioned representations; this specific instantiation has new elements, but it seems overstated to me, and these other works are not really discussed. Taskonomy, Task2Vec, many works in robotics/RL, FiLM, come to mind.
- It's unclear how realistic the self-made tasks are relative to established similar tasks.
- Missing discussion of other work that uses LLMs to generate descriptors for further use, such as https://arxiv.org/abs/2209.03320 / https://arxiv.org/abs/2210.07183 / subsequent works building on these


I could be convinced to increase my score with more focused positioning and a compelling reason this setting is worth unique exploration/differentiated from other previously established settings.

---

> ### Author Rebuttal · Authors · 2025-07-31
>
> We are grateful for your constructive and kind comments. Below, we provide the point-by-point response to your concerns.
>
> **Weakness 1 & Question 2: Our contribution**
>
> We clarify that our proposed conditional representation learning is remarkably distinct from works on task-conditioned and goal-conditioned representations. In essence, our method operates at a more fundamental and general level compared to task- or goal-conditioned approaches..
>
> We first make a summary of our work and task/goal-conditioned works as follows:
>
> 1\) Task-conditioned works aim to learns representations that reveal the underlying correlations among different tasks. For example, taskonomy computes the optimal transfer learning paths among tasks (point matching, reshading, etc.) to minimize the amount of required annotation.
>
> 2\) Goal-conditioned works aim to learns representations that meet the required outcomes or goal states. For example, Robotics/RL works typically involve using the goal state as a conditional input to learn policies to instruct the agent from its current state to the goal state.
>
> 3 \) Our work aims to learn representations that adapt to arbitrary user-specified criteria ("color", "texture", etc.), extending the conventional representation learning that focuses on a single dominant aspect (typically “category”).
>
> From the above comparison, we highlight the key difference:
>
> Task-conditioned works focus on generalization across tasks and are commonly used in supervised multi-task learning. Goal-conditioned works, on the other hand, emphasize generalization across goal supervision and are typically applied in reinforcement learning. In contrast, our work targets generalization across user-specified criteria, which is situated in the context of unsupervised representation learning, a more fundamental and upstream field.
>
> We will clarify this positioning more clearly in the introduction section.
>
> **Weakness 2 & Question 3: Task realism (Framing existing tasks under this setting)**
>
> The proposed concept of conditional representation learning holds significant promise and research merit. Existing representation learning works primarily focus on learning a representation under a universal criterion ("shape" or "category"). However, this may not always align with customized downstream tasks. For instance, in animal habitat analysis, researchers prioritize scene-related features, whereas universal embeddings emphasize categorical semantics.  Moreover, in the fashion retrieval scenario, users often search for items based on different criteria, such as color, material, or occasion. A user might look for "red dresses" today, "formal outfits" tomorrow, and "lace tops" the next day. While the input image remains the same, the relevant features for retrieval vary depending on the user's intent. Conditional representation learning enables generating representations that are tailored to the given criterion conveniently and quickly, improving user satisfaction.
>
> We propose conditional representation learning and offer an efficient and generalizable solution. We utilize it as a plug-and-play module to apply to four downstream tasks: linear probe  (Section 4.1.1), clustering (Section 4.1.2), customized similarity retrieval (Section 4.2.1), and fashion retrieval (Section 4.2.2). In fact, they are not self-made tasks. To be specific, these four tasks are derived from classification and retrieval, the most fundamental tasks to evaluate representation learning.
>
> **Weakness 3: Discussions on LLM-based descriptor generation works**
>
> The first paper you mentioned (paper 1) belongs to prompt learning, and we discuss these works in Question 6.
>
> The second paper (paper 2) in the comment is similar to LaBo and LM4CV in our Related Work section (line 109), belonging to interpretable classification.
>
> Compared to these two papers, CRL differs in the following two aspects:
>
> 1\) Generated content: They use LLM to generate content to describe each class name, implicitly related to universal "category" criterion. For example, paper 1 asks the LLM to generate "two legs", "a small head", etc., for the class "hen". In comparison, our work uses LLM to generate content to describe the criterion, which is a higher-level concept. The generated results can be understood as class names, which can be found in our appendix.
>
> 2\) Purpose: Paper 1 aims to improve classification accuracy and interpretability under the universal criterion. Paper 2 aims to learn the optimal prompt to improve zero-shot image classification accuracy. In contrast, our work focuses on obtaining conditional representations across different criteria.
>
> **Question 1: Comparison to text-conditioned representation methods**
>
> The baselines (Combiner, LinCIR, etc.)  in Section 4.2.1 are exactly the methods you mentioned. These methods obtain image representations conditioned on text already, and then use the new representations to retrieve the most similar images from the gallery.
>
> Compared to these methods, CRL does not require task-specific design and can be used in a plug-and-play manner. Nevertheless, CRL needs to be constructed on models where image and text representations are made separately, thus it’s hard to apply CRL to backbones that are not inherently designed for this type of architecture.
>
> By the way, we apply CRL to BLIP2 and achieve consistently ideal performance improvement like CLIP, which further demonstrates the generalizability of CRL.
>
> | Task |CLIP|CLIP+CRL|BLIP2|BLIP2+CRL|
> | :----------: | :--------: | :--------: | :--------: | :--------: |
> |  Clustering  | 21.73±0.28 | 44.96±0.52 | 26.07±0.79 | 51.64±0.64 |
> | Linear Probe | 43.93±0.43 | 53.34±0.44 | 49.78±0.61 | 59.58±0.35 |
>
> **Question 4: With how many examples do the improvements level off**
>
> Firstly, we would like to clarify that CRL itself is unsupervised and does not rely on additional labeled examples. When applying to downstream tasks, the training data we use is the same as the corresponding baselines.
>
> We understand that you might be referring to the number of texts generated by the LLM. We have explored this in Section 4.3. As Figure 6 indicates, when the number of generated texts increases to around 100, the performance improvements level off across all four tasks.
>
> **Question 5.1: Input for the evaluated tasks**
>
> The input for the evaluated tasks includes the criterion and the LLM/VLM prompt.
>
> **Question 5.2: Method Robustness**
>
> We now present an analysis of CRL's robustness for these factors.
>
> **Robustness against different LLMs**
>
> As for the selection of LLMs, we would like to clarify that our method does not particularly rely on any specific LLM. To be specific, we use the same prompt to query four mainstream LLMs, GPT-4o, Deepseek-v3, Gemini 2.5, and Claude 4.
>
> |     Task     |   GPT-4o   | Deepseek-v3 |  Gemini 2.5  |   Claude 4   | Std  |
> | :----------: | :--------: | :---------: | :----------: | :----------: | :--: |
> |  Clustering  | 44.96±0.52 | 43.40±0.50  | 43.80 ± 0.55 | 43.75 ± 0.58 | 0.59 |
> | Linear Probe | 53.34±0.44 | 52.94±0.40  | 52.93 ± 0.41 | 53.43 ± 0.45 | 0.23 |
>
> The result demonstrates that CRL remains stable when adopting different LLMs to generate the texts.
>
> **Robustness against different LLM prompts**
>
> As for the LLM prompt, we devise 5 different templates:
>
> 1\) Generate common expressions to describe the [criterion].
>
> 2\) List a wide variety of typical phrases used to characterize the [criterion].
>
> 3\) Enumerate familiar terms or expressions people often use when referring to the [criterion].
>
> 4\) Identify and list expressions frequently used to convey the concept of the [criterion].
>
> 5\) How do people usually talk about the [criterion]?
>
> |Task|1|2|3|4|5|Std|
> | :----------: | :--------: | :--------: | :--------: | :--------: | :--------: | :--: |
> |Clustering| 44.96±0.52 | 42.45±0.44 | 42.87±0.31 | 44.75±0.65 | 42.60±0.55 | 1.10 |
> |Linear Probe | 53.34±0.44 | 52.95±0.39 | 53.42±0.37 | 52.82±0.48 | 52.40±0.44 | 0.37 |
>
> The table above suggests that CRL is robust against the LLM prompt.
>
> **Robustness against different VLM prompts**
>
> As for the VLM prompt, we also devise 5 different templates:
>
> 1\) objects with the [criterion] of [text]
>
> 2\) a photo with the [criterion] of [text]
>
> 3\) itap with the [criterion] of [text]
>
> 4\) art with the [criterion] of [text]
>
> 5\) a cartoon with the [criterion] of [text]
>
> |Task|1|2|3|4|5|Std|
> | :----------: | :--------: | :--------: | :--------: | :--------: | :--------: | :--: |
> |Clustering|44.96±0.52|43.72±0.44|44.78±0.46|43.07±0.53|42.33±0.49|1.00|
> |Linear Probe|53.34±0.44|52.28±0.38|52.56±0.42|53.48±0.38|52.58±0.23|0.47|
>
> As suggested in the above table, CRL achieves stable performance across different VLM prompts.
>
> **Question 6: Comparison to visual prompt learning methods**
>
> We summarize the differences between our work and prompt-learning methods as follows:
>
> 1\) Tasks: prompt learning methods aim to learn an optimal prompt to solve the classification task. By comparison, our method serves as a plug-and-play module for the conditional representation learning task.
>
> 2\) Priors: prompt learning methods rely on the prior of class name, while CRL does not require it. Obtaining such priors is impractical in domains featuring massive classes. For example, in the fashion retrieval task, the DeepFashion dataset consists of 1000 classes. In real-world applications, it's even harder to make class names as priors since they are dynamic.
>
> **Limitations Discussion**
>
> Despite CRL's generalizability across different criteria, it may not outperform clustering methods like CC and SCAN under the universal criterion "shape". This is likely because these methods employ specially targeted designs for clustering under this criterion.
>
> We will add the above results and discussions in the next version.
>
> Finally, we deeply appreciate your time and constructive feedback, and we earnestly hope that our response meets your expectations and resolves the issues raised.

---

> ### Author Response · Authors · 2025-08-04
>
> Dear reviewer YAxP,
>
> We sincerely appreciate the time and effort you have devoted to reviewing our submission. We are writing to kindly follow up on your review comments. We would truly appreciate any further feedback you could provide, as we are eager to incorporate your perspective and improve the quality of our work. Please let us know if there’s anything specific you would like us to address.

---

> > ### Comment · Reviewer_YAxP · 2025-08-05
> >
> > Thank you for the detailed reply to each of the points raised. I lean more positively overall now, with the primarily concerns remaining still being task realism and framing within the relevant works on LLM-generated visual information such as descriptors. I liked the justification for the task setting given in the reply here. By self-made I meant the particular evaluations done, not retrieval and classification, of course; this concern remains. Given additional discussion of the framing, I would raise my score to borderline accept, but not higher given the remaining concerns of task realism.

---

> > > ### Author Response · Authors · 2025-08-05
> > >
> > > Thank you for your positive and timely feedback. We are delighted to see that our response has addressed some of your concerns.
> > >
> > > As for the task realism, we apologize for not clarifying it clearly in the previous response. In fact, except for the customized linear probe, the other 3 tasks — customized clustering, customized similarity retrieval, and customized fashion retrieval — are all existing benchmarks. Moreover, in addition to the original CLIP representations, we further included comparisons with task-specific baselines tailored for corresponding benchmarks.
> > >
> > > Specifically, the customized clustering task (section 4.1.2) was proposed by [1, 2]. Among the baselines, [3, 4] are representative methods from the general clustering literature, while [5] was originally proposed for customized clustering.
> > >
> > > The customized similarity retrieval task (section 4.2.1) was proposed by [6], with baselines [7, 8, 9, 10, 11] widely used for this task.
> > >
> > > The customized fashion retrieval task (section 4.2.2) was proposed by [12], with [13, 14, 15] serving as baseline methods specifically developed for this task.
> > >
> > > In other words, the tasks in our work — except for the customized linear probe — are established benchmarks rather than self-made tasks.
> > >
> > > Finally, thanks so much for your time and effort, and we hope our response could address your concerns.
> > >
> > >
> > >
> > > [1] Bae, Eric, and James Bailey. "Coala: A novel approach for the extraction of an alternate clustering of high quality and high dissimilarity." *Sixth International Conference on Data Mining (ICDM'06)*. IEEE, 2006.
> > >
> > > [2] Gondek, David, and Thomas Hofmann. "Non-redundant data clustering." *Knowledge and Information Systems* 12.1 (2007): 1-24.
> > >
> > > [3] Li, Yunfan, et al. "Contrastive clustering." *Proceedings of the AAAI conference on artificial intelligence*. Vol. 35. No. 10. 2021.
> > >
> > > [4] Van Gansbeke, Wouter, et al. "Scan: Learning to classify images without labels." *European conference on computer vision*. Cham: Springer International Publishing, 2020.
> > >
> > > [5] Yao, Jiawei, Qi Qian, and Juhua Hu. "Multi-modal proxy learning towards personalized visual multiple clustering." *Proceedings of the IEEE/CVF Conference on Computer Vision and Pattern Recognition*. 2024.
> > >
> > > [6] Vaze, Sagar, Nicolas Carion, and Ishan Misra. "Genecis: A benchmark for general conditional image similarity." *Proceedings of the IEEE/CVF Conference on Computer Vision and Pattern Recognition*. 2023.
> > >
> > > [7] Saito, Kuniaki, et al. "Pic2word: Mapping pictures to words for zero-shot composed image retrieval." *Proceedings of the IEEE/CVF Conference on Computer Vision and Pattern Recognition*. 2023.
> > >
> > > [8] Baldrati, Alberto, et al. "Zero-shot composed image retrieval with textual inversion." *Proceedings of the IEEE/CVF International Conference on Computer Vision*. 2023.
> > >
> > > [9] Gu, Geonmo, et al. "Language-only training of zero-shot composed image retrieval." *Proceedings of the IEEE/CVF Conference on Computer Vision and Pattern Recognition*. 2024.
> > >
> > > [10] Wang, Lan, et al. "Generative zero-shot composed image retrieval." *Proceedings of the Computer Vision and Pattern Recognition Conference*. 2025.
> > >
> > > [11] Baldrati, Alberto, et al. "Conditioned and composed image retrieval combining and partially fine-tuning clip-based features." *Proceedings of the IEEE/CVF Conference on Computer Vision and Pattern Recognition*. 2022.
> > >
> > > [12] Veit, Andreas, Serge Belongie, and Theofanis Karaletsos. "Conditional similarity networks." *Proceedings of the IEEE conference on computer vision and pattern recognition*. 2017
> > >
> > > [13] Ma, Zhe, et al. "Fine-grained fashion similarity learning by attribute-specific embedding network." *Proceedings of the AAAI Conference on artificial intelligence*. Vol. 34. No. 07. 2020.
> > >
> > > [14] Dong, Jianfeng, et al. "Fine-grained fashion similarity prediction by attribute-specific embedding learning." *IEEE Transactions on Image Processing* 30 (2021): 8410-8425.
> > >
> > > [15] Dong, Jianfeng, et al. "From region to patch: Attribute-aware foreground-background contrastive learning for fine-grained fashion retrieval." *Proceedings of the 46th International ACM SIGIR Conference on Research and Development in Information Retrieval*. 2023.

---

### Official Review · Reviewer_xrw3 · 2025-07-03

**Clarity:** 3
**Significance:** 2
**Originality:** 2
**Rating:** 4
**Confidence:** 4

**Summary:**

This manuscript tackles to learn multiple representation that can capture various semantics on given conditions. For a given criterion like color, it generate descriptive texts semantically related to criterion by using LLM. Then it encodes generated texts and images through VLM like CLIP to generate conditional representation by projecting image representation into representation space of generated texts. Experiments on Clevr4-10k and Card benchmarks demonstrate the effectiveness of the proposed method.

**Questions:**

See the above Strengths And Weaknesses section.

**Ethical Concerns:**

["NO or VERY MINOR ethics concerns only"]

**Final Justification:**

Some of my concerns regarding additional baselines, such as BLIP2, have been addressed. However, as reported in Table 2 of the manuscript, the proposed method exhibits substantial performance gaps in certain tasks, such as the color task. This raises questions about whether the method can consistently improve performance across all tasks.

With regard to technical novelty, I am not convinced by the provided rebuttal. The proposed approach seems largely similar to the original CLIP method. This is particularly evident in the previously mentioned observation that the original CLIP often outperforms the proposed + CRL method on several metrics, including Shape, Number, and Suits. So, the key contribution that generating predefined classes for zero-shot classification using an LLM (instead of humans) sounds incremental to me.

Additionally, this manuscript does not provide new benchmark results like VQA yet.

During the discussion period, the authors addressed the remaining concerns further. I raised my score to 4, but it is still close to borderline.

**Limitations:**

No. I can not find explicit limitations and social impact sections in this manuscript. Although the authors mentioned that they discuss the limitations in the Experiment section, I can not find them in their section 4.

**Paper Formatting Concerns:**

This manuscript does not have explicit limitation and social impact sections.

**Quality:**

2

**Strengths And Weaknesses:**

- Learning conditional representation is interesting and important.
- The proposed method is quite simple and technical novelty is limited. The equation 4 is equivalent conventional logits for zero-shot classification of CLIP. The proposed method uses LLM to class categories and produces zero-shot logits on given classes, and then employ the logits as the conditional representation. Empirical results in Table 1 also show that the original CLIP often outperforms the proposed method CLIP + CRL on various metrics like Shape, Number, and Suits.
- This manuscript only performs the evaluation on a few benchmarks. As I think conditional representation is closely related to VQA problem, I suggest to include evaluations on common vqa benchmarks like COCO, VQAv2, and so on.
- Only CLIP based baselines are compared to this work. For example, BLIP2 Qformer [1] could also produce image representation conditioned on given texts. More comparisons and discussions to explain how can the proposed simple method work better than the existing vision-language approaches.

[1] BLIP-2: Bootstrapping Language-Image Pre-training with Frozen Image Encoders and Large Language Models,

---

> ### Author Rebuttal · Authors · 2025-07-31
>
> We appreciate your careful comments. However, we believe that the judgment may overlook key aspects of our contribution, which we would like to clarify below.
>
> **Weakness 1.1: Technical novelty**
>
> The concept of conditional representation learning we propose is of high potential and research value. Existing representation learning works primarily focus on learning a representation under a universal criterion ("shape" or "category"). However, this may not always align with customized downstream tasks. For instance, in animal habitat analysis, researchers prioritize scene-related features, whereas universal embeddings emphasize categorical semantics.  Moreover, in the fashion retrieval scenario (Section 4.2.2), users often search for items based on different criteria, such as color, material, or occasion. A user might look for "red dresses" today, "formal outfits" tomorrow, and "lace tops" the next day. While the input image remains the same, the relevant features for retrieval vary depending on the user's intent. Conditional representation learning enables the model to generate image representations that are tailored to the given criterion conveniently and quickly, improving retrieval efficiency and user satisfaction.
>
> We propose conditional representation learning and offer a computationally efficient and highly generalizable solution. This is exactly where our novelty lies.
>
> Besides, our proposed method is a simple, plug-and-play approach. Nevertheless, we believe this is not a weakness but rather a strength. Furthermore, we provide a basis transformation perspective to justify the technical soundness of our method.
>
> **Weakness 1.2: Comparison to zero-shot classification**
>
> While equation 4 is similar in computational form to zero-shot classification, our method is fundamentally different in the following two aspects:
>
> 1\) Different tasks: Zero-shot classification directly outputs a predicted class label, solving the classification task. By comparison, our method serves as a plug-and-play module for the conditional representation learning task.
>
> 2\) Different priors: Zero-shot classification relies on the availability of class names as prior knowledge, while our method does not require them. Obtaining such priors is impractical in domains featuring a large number of classes. For example, in the fashion retrieval task(Section 4.2.2), the DeepFashion dataset we adopt consists of 1000 fine-grained classes. What's more, in real-world applications, it's even harder to make class names as priors since they are dynamic.
>
> **Weakness 1.3: Performance**
>
> We respectfully note that the performance degradation does not occur as "often" as the comment suggested.
>
> In fact, across all four tasks, our method consistently outperforms the base model and existing baselines. Specifically, for the linear probe task (Table 1), CRL obtains a performance boost of the original CLIP representation of over 21% relative. For the clustering task (Table 2), CRL obtains a performance boost of the SOTA method (Multi-Map) of over 53%. For the customized similarity retrieval task (Table 3), CRL achieves a gain of 13% compared to the SOTA method (Combiner). For the fashion retrieval task (Table 4), CRL surpasses the best competitive method (RPF) by 10%. The few cases of degradation mentioned in the comment are isolated exceptions rather than the norm.
>
> **Weakness 2: Benchmark**
>
> Our work falls under the scope of representation learning. Therefore, we selected downstream tasks that are closely tied to this area, specifically, classification and retrieval. We included four such sub-tasks, which we believe are sufficient to validate the effectiveness of our approach.
>
> Nevertheless, we truly value the suggestion of applying our method to the VQA problem. Since we are not familiar with this field and previous representation learning works seldom select the VQA problem as the downstream task, we wonder how to apply our method to it.
>
> By the way, we believe this is feasible, as the dataset used in the customized similarity retrieval task (Section 4.2.1) is derived from the COCO dataset. We'll leave this as a promising direction for future exploration.
>
> **Weakness 3: Baseline**
>
> Thanks for the insightful suggestion. Before the rebuttal, we indeed only considered applying our method based on the CLIP-like architecture. Following your suggestion, we apply our method to BLIP2 and find that it even outperforms CLIP, as the following table shows. This further demonstrates the generalizability of our work, and we will continue exploring broader applicability in future research.
>
> |     Task     |     CLIP     |       CLIP + CRL       |   BLIP2    |     BLIP2 + CRL      |
> | :----------: | :----------: | :--------------------: | :--------: | :------------------: |
> |  Clustering  | 21.73 ± 0.28 | 44.96 ± 0.52 (↑23.23%) | 26.07±0.79 | 51.64±0.64 (↑25.57%) |
> | Linear Probe | 43.93 ± 0.43 | 53.34 ± 0.44 (↑9.42%)  | 49.78±0.61 | 59.58±0.35 (↑9.80%)  |
>
> Moreover, except for the linear probe task, the other three tasks are also compared against methods specifically designed for those tasks. To be specific, for the clustering task (Section 4.1.2), the baselines include three specifically designed methods: CC, SCAN, and Multi-Map. For the customized similarity retrieval task (Section 4.2.1), the baselines include five representative methods: Pic2Word, SEARLE, LinCIR, CIG, and Combiner. For the fashion retrieval task (Section 4.2.2), we compare against five competitive baselines: Triplet, CSN, ASEN, ASEN++, and RPF.
>
> **Weakness 4: Discussion on limitations**
>
> Despite CRL's generalizability across different criteria, it may not outperform clustering methods like CC and SCAN under the universal criterion "shape". This is likely because these methods employ specially targeted designs for clustering under this criterion.
>
>
>
> We will add the above results and discussions in the next version.
>
> Finally, we would like to thank you for your time and effort once again, and sincerely hope our response could address your concerns.

---

> > ### Comment · Reviewer_xrw3 · 2025-08-06
> >
> > Thank you for your rebuttal. Some of my concerns regarding additional baselines, such as BLIP2, have been addressed. However, as reported in Table 2 of the manuscript, the proposed method exhibits substantial performance gaps in certain tasks, such as the color task. This raises questions about whether the method can consistently improve performance across all tasks.
> >
> > With regard to technical novelty, I am not convinced by the provided rebuttal. The proposed approach seems largely similar to the original CLIP method. This is particularly evident in the previously mentioned observation that the original CLIP often outperforms the proposed + CRL method on several metrics, including Shape, Number, and Suits. So, the key contribution that generating predefined classes for zero-shot classification using an LLM (instead of humans) sounds incremental to me.
> >
> > Additionally, since this manuscript does not provide new benchmark results like VQA yet, I retain my initial score.

---

> > > ### Author Response · Authors · 2025-08-07
> > >
> > > Thank you for the comment.
> > >
> > > However, we would like to kindly point out that the concerns you raised do not appear to align with the content presented in our work. We believe the comment may reflect a misunderstanding of our contributions.
> > >
> > > **Comment 1:** As reported in Table 2 of the manuscript, the proposed method exhibits substantial performance gaps in certain tasks, such as the color task. This raises questions about whether the method can consistently improve performance across all tasks.
> > >
> > > **Response 1:** We've applied our method to 4 tasks — customized linear probe (Table 1), clustering (Table 2), similarity retrieval (Table 3), and fashion retrieval (Table 4). **The results across all four tasks indeed demonstrate that our method surpasses the base CLIP model**, as the table below shows:
> > >
> > > |         Task         | The original CLIP | Ours  | Performance Gain |
> > > | :------------------: | :---------------: | :---: | :--------------: |
> > > |     Linear Probe     |       43.93       | 53.34 |      ↑ 21%       |
> > > |      Clustering      |       21.73       | 44.96 |      ↑ 107%      |
> > > | Similarity Retrieval |       19.90       | 26.80 |      ↑ 35%       |
> > > |  Fashion Retrieval   |       6.08        | 7.93  |      ↑ 30%       |
> > >
> > > Furthermore, we provide the performance improvements for every criterion.
> > >
> > > | Linear Probe | The original CLIP | Ours  | Performance Gain |
> > > | :----------: | :---------------: | :---: | :--------------: |
> > > |   Texture    |       27.70       | 33.28 |      ↑ 20%       |
> > > |    Shape     |       76.93       | 79.19 |       ↑ 3%       |
> > > |    Color     |       51.39       | 82.42 |      ↑ 60%       |
> > > |    Count     |       22.13       | 28.46 |      ↑ 29%       |
> > > |    Number    |       32.07       | 37.76 |      ↑ 18%       |
> > > |    Suits     |       53.33       | 58.97 |      ↑ 11%       |
> > >
> > > | Clustering | The original CLIP | Ours  | Performance Gain |
> > > | :--------: | :---------------: | :---: | :--------------: |
> > > |  Texture   |       4.87        | 14.07 |      ↑ 189%      |
> > > |   Shape    |       70.52       | 78.05 |      ↑ 10%       |
> > > |   Color    |       4.44        | 86.34 |     ↑ 1845%      |
> > > |   Count    |       11.41       | 21.45 |      ↑ 88%       |
> > > |   Number   |       14.73       | 21.71 |      ↑ 47%       |
> > > |   Suits    |       24.40       | 48.15 |      ↑ 97%       |
> > >
> > > **The results indicate that the improvements are not limited to the color criterion. Performances under other criteria have also improved.** Notably, the original CLIP focuses on the Shape criterion, thus the improvement is not as obvious as other criteria.
> > >
> > > **Comment 2:** The proposed approach seems largely similar to the original CLIP method.
> > >
> > > **Response 2:**  **We clarify that the original CLIP method prioritizes a single criterion (Shape), while our method can learn representations tailored to arbitrary criteria.** If you mean the zero-shot, please refer to our previous rebuttal in Weakness 1.2.
> > >
> > > **Comment 3:**  This is particularly evident in the previously mentioned observation that the original CLIP often outperforms the proposed + CRL method on several metrics, including Shape, Number, and Suits.
> > >
> > > **Response 3:** As shown in the above tables and Table 1-4 in our paper, for the mentioned metrics, including Shape, Number, and Suits, our method also surpasses the base CLIP model.
> > >
> > > **Comment 4:** The key contribution that generating predefined classes for zero-shot classification using an LLM (instead of humans) sounds incremental to me.
> > >
> > > **Response 4:** **In fact, our contribution lies in proposing conditional representation learning (towards multiple criteria) and offering an efficient and generalizable solution.** Moreover, we don't generate the predefined classes. As shown in the appendix, we use the LLM to generate massive possible texts under the specified criterion. Zero-shot methods rely on having a complete and accurate prior of all predefined class names. As we responded previously, obtaining such priors is impractical in domains featuring massive and dynamic classes.
> > >
> > > **Comment 5:** This manuscript does not provide new benchmark results like VQA yet.
> > >
> > > **Response 5:** **We must point out that our work focuses on (conditional) representation learning. Classification and retrieval are among the most fundamental downstream tasks for evaluating the learned representation quality.** To thoroughly assess our method, we have conducted experiments on 4 tasks for classification and retrieval. Nevertheless, we are open and delighted to make extensions to our work, but it remains unclear to us how our method could be applied to the VQA task.
> > >
> > > Finally, we appreciate your effort and respectfully understand that some aspects of our work may not have been grasped, possibly due to the time constraints of reviewing. We hope the clarifications below can help better convey our contribution.

---

> > > > ### Comment · Reviewer_xrw3 · 2025-08-07
> > > >
> > > > Thanks for the additional responses. It addressed the remaining concerns further. I raised my score to 4, but it still sounds close to borderline.

---

> > > > > ### Author Response · Authors · 2025-08-08
> > > > >
> > > > > Thanks for the timely response. We will refine our paper following your suggestions.

---

> ### Author Response · Authors · 2025-08-04
>
> Dear reviewer xrw3,
>
> We hope this message finds you well. As the rebuttal deadline is approaching, we wanted to kindly check in to see if you might have any comments or feedback for us to address. Your insights are important to us, and we would like to make sure we have sufficient time to incorporate your suggestions into our response. Thank you very much for your time and effort.

---

### Official Review · Reviewer_iGLB · 2025-07-16

**Clarity:** 2
**Significance:** 2
**Originality:** 2
**Rating:** 4
**Confidence:** 5

**Summary:**

The paper proposes conditional representation learning, a two stage framework for tailoring visual embeddings to a user-specified criterion. In the first step, the authors propose using an LLM to generate texts corresponding to the criteria which is then used as a basis for projecting the VLM embeddings into the same space. The primary motivation is semantic alignment of the embeddings to a task specific domain without fine-tuning. Experimental results are provided on multiple criterion tasks setup on four datasets - Clevr4-10k and Cards for image tasks, and GeneCIS and DeepFashion for retrieval tasks.

**Questions:**

Please refer to the weaknesses section above.

**Ethical Concerns:**

["NO or VERY MINOR ethics concerns only"]

**Final Justification:**

The authors address some of my experimental concerns by providing additional ablations, confidence intervals for results, etc, which partly addresses my original concerns and hence I raised my score. However I still have concerns about the task setup, and missing comparisons with some baseline methods that the authors did not provide during the rebuttal phase, with the primary comparisons being with the base models. Hence, I will do not believe that the submission is strong and convincing enough for a clear accept decision.

**Limitations:**

Yes

**Quality:**

2

**Strengths And Weaknesses:**

## Strengths

1. The key idea proposed by the authors is fairly simple, and the methodology seems straightforward and interpretable.


2. The paper is generally well written and easy to follow.


3. Since the method involves inference on LLMs followed by simple projection operations, it seems to be efficient compute wise.


## Weaknesses

1. There is a large body of work in the domain of semantic alignment in projection spaces, and even domain adaptation/alignment which is missing from either from the related work section, or in use as baselines or even both. TextSpan [1], ICITC [2] are mentioned in the related work, however other relevant works like MMRL [3], SwapPrompt [4], and more are not mentioned. All of them are missing from the use as baselines altogether, hence there is not enough justification of placing their work in the context of related work. In fact, the methodology of TextSpan and ICITC is almost identical to the one proposed by the authors, however there is no discussion comparing and contrasting their work.


2. While the results in experimental section demonstrate strong gains over the unaligned and non-finetuned baselines, the experimental setup is not very convincing. The datasets are on the smaller scale size for both vision and retrieval domains as compared to the real world large scale task datasets. All of the main results in tables 1, 2, 3 and 4 are also missing confidence intervals for statistical significance.

3. Details about the selection of LLM and decision choices about prompts etc are missing. The main paper does not even mention what LLM is used for the generation task, what is the rationale behind the selection, the dependence on GPT-4 and impact of using other LLMs etc. There are also no ablation studies in the paper (or the appendix) to give more intuition into the method and it’s robustness to factors like the prompt, temperature, redundancy in generation and so forth.


4. The advantage that general purpose embeddings offer is the usability across tasks, and hence a comparison with the base models is not a well-formed setup. Plus the use of a naive projection operator can cause issues in correlation across basis vectors, and also cause some degradation in performance. Some of the cases where the performance is degraded, e.g. in Table 1 for clevr4-10k in the “count” task, the failures are not analysed.



[1] INTERPRETING CLIP’S IMAGE REPRESENTATION VIA TEXT-BASED DECOMPOSITION, Gandelsman et. al, ICLR ‘24

[2] IMAGE CLUSTERING CONDITIONED ON TEXT CRITERIA, Kwon et. al, ICLR ‘24

[3] MMRL: Multi-Modal Representation Learning for Vision-Language Models, Guo et. al, CVPR ‘25

[4] SwapPrompt: Test-Time Prompt Adaptation for Vision-Language Models, Ma et. al, NeurIPS ‘23

---

> ### Author Rebuttal · Authors · 2025-07-31
>
> We sincerely thank you for the time and effort spent reviewing our paper. Nevertheless, we respectfully believe that some of the concerns raised may be due to misunderstandings, and we would like to clarify these points in detail below.
>
> **Weakness 1.1: Related work**
>
> First of all, we must point out that the core of our work, CRL (conditional representation learning), aims to obtain conditional representations tailored to the user-specified criterion without any supervision signal.
>
> According to this, we choose "representation learning" and "conditional similarity" as our related work. The former refers to traditional methods that aim to learn general feature representations, while the latter focuses on newly emerging areas that learn conditional similarity conditioned on specific criteria.
>
> The relevant works you mentioned, MMRL (fine-tuning the model structure for image classification) and SwapPrompt (adapting the VLM prompt for image classification), are two works that aim to learn representations under the general criterion (shape or category).  We'll add them to the "representation learning" section in related works.
>
> **Weakness 1.2: Methodology novelty**
>
> We disagree with the comment "the methodology of TextSpan and IC|TC is almost identical to the one proposed by the authors". Below is an intuitive comparison among these three works:
>
> 1\) Different tasks:  CRL aims to obtain conditional representations under specified criteria for multiple tasks. TextSpan seeks to analyze the role of each head in the CLIP model. IC|TC is designed for the clustering task.
>
> 2\) Different techniques: CRL utilizes the descriptive texts as the semantic basis, and then projects images into this conditional feature space to obtain conditional representations. TextSpan iteratively selects a set of texts that maximizes the variance of the head's output features, using these texts to interpret the corresponding head. IC|TC queries the LLM to get the cluster affiliation with three steps.
>
> It's clear to see that CRL is substantially different from TextSpan and IC|TC.
>
> **Weakness 2.1:  Dataset scale**
>
> To the best of our knowledge, previous multi-criteria works typically focused on a single downstream task. So in this paper, we integrate existing multi-criteria datasets from multiple domains.  We acknowledge that, as an emerging field, this area currently lacks abundant high-quality datasets. However, we would like to point out that not all dataset scales are small, as the DeepFashion dataset contains over 270k images (in the 4.2.2 section).
>
> We hope our work can inspire broader interest in this promising research direction. Moving forward, we also intend to develop larger-scale datasets to support continued progress.
>
> **Weakness 2.2: Missing confidence intervals**
>
> We present the confidence intervals (at the confidence level of 0.95) for both the customized clustering and linear probe tasks across 20 trials. The experiments in the rebuttal are all conducted under the same setting.
>
> |     Task     |     CLIP     |    CLIP + CRL(Ours)    |
> | :----------: | :----------: | :--------------------: |
> |  Clustering  | 21.73 ± 0.28 | 44.96 ± 0.52 (↑23.23%) |
> | Linear Probe | 43.93 ± 0.43 | 53.34 ± 0.44 (↑9.42%)  |
>
> The table below reveals that the random seed has minimal impact on the performance of our method.
>
> We'll complement tables 1, 2, 3, and 4 in the main text for the next version.
>
> **Weakness 3.1: LLM ablation**
>
> Thanks for the suggestion, and we provide thorough ablation studies below.
>
> As for the selection of LLMs, we would like to clarify that our method does not particularly rely on any specific LLM. To be specific, we use the same prompt to query four mainstream LLMs, GPT-4o, Deepseek-v3, Gemini 2.5, and Claude 4.
>
> |     Task     |    GPT-4o    | Deepseek-v3  |  Gemini 2.5  |   Claude 4   | Std  |
> | :----------: | :----------: | :----------: | :----------: | :----------: | :--: |
> |  Clustering  | 44.96 ± 0.52 | 43.40 ± 0.50 | 43.80 ± 0.55 | 43.75 ± 0.58 | 0.59 |
> | Linear Probe | 53.34 ± 0.44 | 52.94 ± 0.40 | 52.93 ± 0.41 | 53.43 ± 0.45 | 0.23 |
>
> The result demonstrates that all four LLMs achieve comparable and consistent performance.
>
> **Weakness 3.2: LLM prompt ablation**
>
> As for the LLM prompt, we require it to include the [criterion]. We devise below 5 different templates:
>
> 1\) Generate common expressions to describe the [criterion].
>
> 2\) List a wide variety of typical phrases used to characterize the [criterion].
>
> 3\) Enumerate familiar terms or expressions people often use when referring to the [criterion].
>
> 4\) Identify and list expressions frequently used to convey the concept of the [criterion].
>
> 5\) How do people usually talk about the [criterion]?
>
> |     Task     |   Prompt 1   |   Prompt 2   |   Prompt 3   |   Prompt 4   |   Prompt 5   | Std  |
> | :----------: | :----------: | :----------: | :----------: | :----------: | :----------: | :--: |
> |  Clustering  | 44.96 ± 0.52 | 42.45 ± 0.44 | 42.87 ± 0.31 | 44.75 ± 0.65 | 42.60 ± 0.55 | 1.10 |
> | Linear Probe | 53.34 ± 0.44 | 52.95 ± 0.39 | 53.42 ± 0.37 | 52.82 ± 0.48 | 52.40 ± 0.44 | 0.37 |
>
> The table above shows that different LLM prompts can yield close performance improvement, indicating that our method is robust against the LLM prompt.
>
> **Weakness 3.3: VLM prompt ablation**
>
> As for the VLM prompt, we require it to contain the [criterion] and the generated [text] by the LLM. We also devise below 5 different templates:
>
> 1\) objects with the [criterion] of [text]
>
> 2\) a photo with the [criterion] of [text]
>
> 3\) itap with the [criterion] of [text]
>
> 4\) art with the [criterion] of [text]
>
> 5\) a cartoon with the [criterion] of [text]
>
> |     Task     |   Prompt 1   |   Prompt 2   |   Prompt 3   |   Prompt 4   |   Prompt 5   | Std  |
> | :----------: | :----------: | :----------: | :----------: | :----------: | :----------: | :--: |
> |  Clustering  | 44.96 ± 0.52 | 43.72 ± 0.44 | 44.78 ± 0.46 | 43.07 ± 0.53 | 42.33 ± 0.49 | 1.00 |
> | Linear Probe | 53.34 ± 0.44 | 52.28 ± 0.38 | 52.56 ± 0.42 | 53.48 ± 0.38 | 52.58 ± 0.23 | 0.47 |
>
> As suggested in the above table, our method remains stable across different VLM prompts.
>
> **Weakness 3.4: Temperature ablation**
>
> As for the LLM temperature t, we set t to 0, 0.5, 1 and 1.5 to obtain the text basis, respectively. The temperature ranges from 0 to 2, with higher values introducing more variability and randomness in the LLM's output (When the temperature approaches 2, the generated content becomes almost entirely random, so we did not include this setting in our experiments).
>
> |     Task     |     t=0      |    t=0.5     |     t=1      |    t=1.5     | Std  |
> | :----------: | :----------: | :----------: | :----------: | :----------: | :--: |
> |  Clustering  | 43.33 ± 0.73 | 43.74 ± 0.66 | 44.96 ± 0.52 | 43.27 ± 0.39 | 0.68 |
> | Linear Probe | 53.07 ± 0.49 | 53.27 ± 0.48 | 53.34 ± 0.44 | 52.50 ± 0.41 | 0.33 |
>
> The experimental results also validate the robustness of our method to the temperature parameter.
>
> **Weakness 3.5: Redundancy ablation**
>
> As for the redundancy, we analyze the number of generated words in Figure 6 in the main text. The result shows that our method maintains stable performance, indicating robustness to redundancy.
>
> **Weakness 4.1:  Baseline**
>
> It appears that the scope of our baseline comparisons may have been overlooked. Our proposed method is built upon the representations of base models. Consequently, it is intuitive to observe the effect of representation transformation when compared directly to these representations.
>
> However, we don't limit our comparisons to the base models. In fact, except for the linear probe task (which is a direct evaluation of the transformation), the other three tasks are also compared against methods specifically designed for those tasks.
>
> To be specific, for the clustering task (Section 4.1.2), the baselines include three specifically designed methods: CC, SCAN, and Multi-Map. For the customized similarity retrieval task (Section 4.2.1), the baselines include five representative methods: Pic2Word, SEARLE, LinCIR, CIG, and Combiner. For the fashion retrieval task (Section 4.2.2), we compare against five competitive baselines: Triplet, CSN, ASEN, ASEN++, and RPF.
>
> **Weakness 4.2 Performance**
>
> We think this comment may be overly strict. In fact, our method, CRL,  shows significant improvements over both the base models and existing baselines across all four tasks. Specifically, for the linear probe task, CRL obtains a performance boost of the original CLIP representation of over 21% relative. For the clustering task, CRL obtains a performance boost of the SOTA method (Multi-Map) of over 53%. For the customized similarity retrieval task, CRL achieves a gain of 13% compared to the SOTA method (Combiner). For the fashion retrieval task, CRL surpasses the best competitive method (RPF) by 10%.In other words, the cases of performance degradation are isolated exceptions rather than indicative of overall trends. Moreover, we would like to clarify that for the “count” criterion on the clevr4-10k dataset in Table 1, there is no actual performance degradation.
>
>
>
> We will add the above results and discussions in the next version.
>
> Finally, we sincerely hope that our response will be carefully considered and lead to a more favorable assessment. Thank you for the time spent reviewing our paper again.

---

> ### Author Response · Authors · 2025-08-04
>
> Dear reviewer iGLB,
>
> We are reaching out with a gentle reminder as the rebuttal submission deadline is drawing near. We noticed that your comments have not yet been posted, and we would be very grateful if you could share any feedback or concerns in the remaining time. Your input is crucial for us to respond meaningfully. We sincerely appreciate your efforts and look forward to your thoughts.

---

### Note · Authors · 2025-08-13

Dear Area Chair and reviewers,

We hope everything goes well.

In the last period, we would like to summarize our rebuttal in the following three points:

1. **Novelty.** In contrast to traditional representation learning works that focus on learning a universal representation based on a single criterion, we propose conditional representation learning towards arbitrary criteria for multiple downstream tasks. This highlights a key distinction between our work and prior studies.
2. **Performance.** We offer an efficient and generalizable solution to achieve conditional representation learning. Across four tasks — customized linear probe, clustering, similarity retrieval, and fashion retrieval, our method consistently outperforms the base model and existing methods tailored for these tasks.
3. **Robustness.** We conduct extensive ablation experiments on LLM, LLM prompt, VLM prompt, temperature, and redundancy. We also replace CLIP-like architectures with BLIP2 as the backbone. The results demonstrate that our method is robust against these variants.

We hope this work could inspire new insights and stimulate further research in conditional representation learning, an underexplored yet promising field we propose.

Finally, we sincerely thank the Area Chair and all reviewers for their time, efforts, and constructive feedbacks, which have helped us to improve the paper.

Best wishes,

Authors of Paper 16397

---

### Decision · Program_Chairs · 2025-09-17

**Decision:**

Accept (spotlight)

**Comment:**

This paper deals with a drawback of the existing class of vision-language models, where such models strive for a universal representation of everything, leading to limited performance in fine-grained and specialized tasks. This paper introduces conditional representation learning, in which representation are extracted to match any user-specified criterion. After the review and rebuttal stage, the reviews unanimously voted above the acceptance threshold (5, 4, 4, 4). The reviewers are clear and short regarding the strengths: the idea is simple yet effective. The AC agrees and finds that the simplicity of the approach is not a valid argument to argue for lack of technical novelty. The reviewers do point out valid concerns regarding comparisons and further experiments. The authors have amongst other included BLIP-2 comparisons and more linear probing evaluations, strengthening the experiments. Based on these outcomes, the AC follows the review recommendations and votes for acceptance.